# Evolving Graph Structured Programs for Circuit Generation with Large Language Models

**Yinqi Bai**[1], **Jie Wang**[1][†], **Lei Chen**[2], **Zhihai Wang**[1], **Yumeng Li**[1], **Mingxuan Yuan**[2],
**Jianye Hao**[3], **Defu Lian**[1], **Enhong Chen**[1],
[1]MoE Key Laboratory of Brain-inspired Intelligent Perception and Cognition,
University of Science and Technology of China
[2]Huawei Technologies Co., Ltd.
[3]College of Intelligence and Computing, Tianjin University
byq000324@mail.ustc.edu.cn,

## Abstract

Logic synthesis (LS), which aims to generate a *compact* logic circuit graph with minimized size while *accurately* satisfying a given functionality, plays an important role in chip design. However, existing LS methods struggle to balance circuit structure compactness and functional accuracy, often leading to suboptimal generation. To address this problem, we propose a novel *Circuit Program Evolution* framework, namely CircuitEvo, which iteratively leverages the large language models (LLMs) to evolve circuit programs towards improved compactness while preserving functional accuracy. Specifically, CircuitEvo models the circuit graph as a structured program and leverages the strong generative capabilities of LLMs—guided by the domain-specific evolutionary prompt strategies—to generate promising circuit candidates in each iteration. Moreover, a structure-aware circuit optimization module is introduced to correct functional discrepancies by appending necessary substructures to the generated circuits. To the best of our knowledge, CircuitEvo is *the first* LLM-based LS approach that can iteratively improve the circuit's compactness while ensuring functional accuracy. Experiments on several widely used benchmarks demonstrate that our CircuitEvo can efficiently generate accurate circuits with up to 16 input number and 69 output number. Moreover, our method significantly outperforms state-of-the-art methods in terms of the circuit size, achieving an average improvement of 6.74%. The code is available at https://github.com/MIRALab-USTC/CircuitEvo.

## 1 Introduction

Complex integrated circuits (ICs) can contain billions of transistors, making manual design impractical (Huang et al., 2021). Consequently, the IC industry depends on electronic design automation (EDA) tools (Wang et al., 2009), which systematically transform high-level hardware descriptions into layouts ready for IC fabrication. A crucial step in this process is logic synthesis (LS) (Devadas et al., 1994; Rudell, 1989), which aims to transform a behavioral-level description, i.e., truth table, into an optimized gate-level circuit to minimize its delay and area. Since LS is the first step in the EDA process that leads to the final IC layout, the quality of its output has a significant impact on the area, power, and performance of the resulting IC (De Abreu et al., 2021; Berndt et al., 2022).

LS is a challenging $\mathcal{NP}$-hard combinatorial optimization problem (Micheli, 1994; Farrahi & Sarrafzadeh, 1994). To address this task, existing LS approaches typically follow a generate-then-optimize framework: they first generate a circuit from the truth table, then apply various operators for circuit optimization (Wang et al., 2024b; Li et al., 2025; Cheng et al., 2024). Since the quality of the initial generated circuit heavily influences the final result after optimization (Rai et al., 2021; Alanminko, 2024), the central problem in LS lies in generating a compact circuit graph that exactly satisfies the functionality constraints defined by the truth table. Both commercial and academic LS tools use sophisticated human-designed heuristics (Nabulsi et al., 2017; Lai et al., 1993) to solve

---

[†]Corresponding author. Email: jiewangx@ustc.edu.cn.

this task, often resulting in suboptimal solutions. Recently, learning-based approaches (Wang et al., 2024b; d'Ascoli et al., 2023; Rai et al., 2021; Li et al., 2025) have emerged as a promising alternative for circuit generation. These methods aim to explore the circuit design space to identify graph structures that meet strict function constraints. However, they struggle to balance circuit structure compactness and functional accuracy, often resulting in limited circuit quality.

To address this challenge, we propose a novel *Circuit Program Evolution* framework, namely CircuitEvo, which iteratively leverages large language models (LLMs) to evolve circuit programs toward improved compactness while preserving functional accuracy. Specifically, CircuitEvo first introduces an innovative graph-structured circuit program formulation (see Figure 1) to model the circuit graph as an LLM-friendly textual format. The key advantage of the circuit program over traditional sequence-based textual representation (d'Ascoli et al., 2023; Li et al., 2025) lies in its graph-structured design, which effectively encodes hierarchical structural information to facilitate LLM's graph comprehension and generation. To effectively explore the complex program space for accurate and compact circuits, CircuitEvo integrates an evolutionary program generator and a structure-aware function optimizer into an iterative framework. In each iteration, CircuitEvo utilizes an LLM-based generator to generate diverse and promising circuit programs. The key innovation of this generator is a domain-specific evolutionary prompting strategy that combines domain knowledge with the explorative power of evolutionary methods to enhance the LLM's generative capabilities.

However, due to the stochastic nature of evolutionary generation and the inherent limitations of LLMs in exactly understanding graph functionality (Dai et al., 2024), the generated circuits cannot always guarantee functional correctness. To address this problem, we introduce a structure-aware function optimizer. Specifically, the optimizer utilizes existing LS tools to efficiently generate two essential substructures, which are then appended to the incomplete circuit to correct functional discrepancies based on the circuit functionality completion theory. Moreover, to ensure the compactness of the optimized circuits, we introduce a local program search method to identify key structures of the generated circuits before functional optimization. By integrating the structure generator with the function optimizer, CircuitEvo effectively explores the space of circuit programs, enabling the generation of both functionally accurate and compact circuits.

We evaluate CircuitEvo using two open-source models——Deepseek-V3 and Qwen2.5-7B-Instruct, and a proprietary model——GPT-3.5-turbo as the backbone large language model. Experiments on several widely used benchmarks demonstrate that CircuitEvo can efficiently generate accurate circuits with up to 16 input number and 69 output number, and significantly achieves an average improvement of 6.74% in terms of the circuit size over state-of-the-art methods. Additionally, through a thorough analysis of CircuitEvo, we observed two key factors contributing to circuit compactness, which can offer valuable insights for the development of future circuit generation approaches.

We summarize our contributions as follows. (1) To the best of our knowledge, CircuitEvo is *the first* LS method that integrates LLM into an evolutionary framework to iteratively improve the circuit's compactness while ensuring functional accuracy. (2) We propose a novel graph-structured circuit program formulation that enables large language models to effectively understand and evolve circuit graphs. (3) Experiments on several widely used benchmarks demonstrate that CircuitEvo can efficiently generate accurate circuits and significantly outperforms the state-of-the-art methods in terms of the circuit size. (4) The thorough analysis of our CircuitEvo provides valuable insights into the key factors that contribute to achieving compact circuits.

## 2 BACKGROUND

**Logic Synthesis (LS) from IO examples** Logic synthesis aims to convert a behavioral-level description, i.e., truth table, into an optimized gate-level circuit as shown in Figure 5 in Appendix B. A truth table is a tabular data in which each row corresponds to an input-output pair $(\mathbf{x}_i, \mathbf{y}_i)$. Given a truth table $\mathcal{T}$ with $n$ inputs and $m$ outputs, the set of input-output pairs $\mathcal{D} = (\mathbf{X}, \mathbf{Y}) = \{(\mathbf{x}_i, \mathbf{y}_i)\}_{i=1}^{2^n}$ defines a Boolean function $f : \mathbb{B}^n \to \mathbb{B}^m$, where $\mathbb{B} = \{0, 1\}$. Thus, the goal of LS is to find a circuit with minimized size while accurately satisfying the given Boolean function $f$. However, directly generating the optimal circuit is challenging due to its $\mathcal{NP}$-hard nature (Farrahi & Sarrafzadeh, 1994). Consequently, LS is divided into two stages—Circuit Generation and Circuit Optimization. The Circuit Generation stage focuses on generating an initial circuit that accurately implements the given truth table. Since the quality of the initial circuit has a significant impact on the quality of the optimized circuits, this stage aims to generate circuits that are both functionally correct and

structurally compact. Following this, the Circuit Optimization stage applies subsequent optimization operators (Mishchenko et al., 2011; Brayton, 2006) to further improve circuit quality.

## 3 RELATED WORK

**Machine Learning for Logic Synthesis (LS)** Traditional LS methods synthesize a circuit from a given truth table via manually designed heuristics, such as sum-of-products (SOP) (Nabulsi et al., 2017) and binary decision diagrams (BDD) (Lai et al., 1993). Recently, many researchers have investigated machine learning for LS (Rai et al., 2021; Belcak & Wattenhofer, 2022a; Schmitt et al., 2021; 2023; Wang et al., 2024a; Bai et al., 2025b; 2026), which offers promising approaches to generating compact circuits with smaller sizes. Existing learning-based approaches can be broadly classified into two categories based on their underlying circuit representations. (1) In the early stages of the International Workshop on Logic & Synthesis (IWLS) competition (Rai et al., 2021), researchers represent the circuit as sequence-based or tree-based Boolean expressions and formulate LS as a symbolic regression problem, i.e., learning accurate and compact Boolean functions given the input-output pairs. Within this formulation, a variety of symbolic regression techniques (Rai et al., 2021; d'Ascoli et al., 2023) have been explored for Boolean function discovery. (2) In recent IWLS competitions (Alanminko, 2024), researchers represent the circuit as a logic gate network, such as And-Inverter Graphs (AIGs), and have proposed using deep machine learning methods (Wang et al., 2024b; Li et al., 2025) to generate circuits.

**Large Language Models for Algorithm Design** In recent years, the capabilities of large language models (LLMs) have advanced substantially (Liang et al., 2025; 2026; Ni et al., 2026; **?**). A growing body of research has explored the use of LLMs as automated algorithm designers, capable of generating novel algorithms through in-context learning. Given that circuit generation can be framed as an algorithm design task, where each circuit represents a functionally constrained graph-based algorithm, we propose to leverage LLMs for the discovery of circuit algorithms. Recent studies have adopted a program-centric approach, representing algorithms as programs and employing LLMs to iteratively optimize them (Romera-Paredes et al., 2024; Liu et al., 2024b). This iterative program generation and refinement process presents a promising direction for discovering new and effective algorithms.

## 4 THE CIRCUITEVO FRAMEWORK

In this section, we provide a comprehensive description of the proposed Circuit Program Evolution framework (CircuitEvo), as illustrated in Figure 2. The structure of the section is as follows: in Section 4.1, we introduce the novel graph-structured circuit program representation and problem formulation. Then, we introduce the two main components of CircuitEvo: Evolutionary Program Generation in Section 4.2, and Structure-aware Function Optimization in Section 4.3.

### 4.1 THE CIRCUIT PROGRAM REPRESENTATION AND PROBLEM FORMULATION

**Graph Structured Circuit Program** Existing works usually employ expression trees (d'Ascoli et al., 2023; Bai et al., 2025a; Sun et al., 2022) and And-Inverter Graphs (Li et al., 2025; Wang et al., 2024b) to model the circuit. However, these non-textual representations present significant challenges for LLM-based understanding and generation. To address this problem, we propose an innovative graph-structured circuit program, which encodes both the circuit's topological structure and functional information into a textual representation as shown in Figure 1. Specifically, the program consists of three parts:

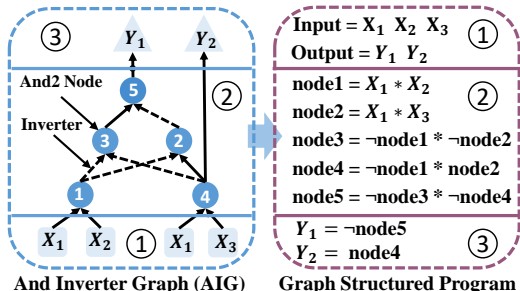

Figure 1: An Illustration of our novel graph-structured circuit program formulation.

1. **IO Definition:** Specify the name and number of input and output variables for the circuit.

2. **Structure Description:** The topological structure of a circuit is inherently defined by its nodes and the directed edges that connect them. In our program representation, we employ a hierarchical, bottom-up modeling approach, where each line specifies a circuit node along with its input dependencies. This formulation explicitly encodes the connectivity among nodes in a textual format, facilitating LLM's comprehension of the circuit's structure.

3. **Function Definition:** Since our program explicitly defines the connectivity between nodes, the Boolean function of each node can be systematically constructed. Therefore, we define the functionality of our circuit program as the Boolean functions of the primary output nodes, which are required to satisfy the specified truth table.

Overall, our graph-structured circuit program formulation bridges the gap between circuit graph representations and language-based reasoning, thereby unlocking the significant potential of LLMs for addressing LS tasks through their powerful generative capabilities.

**Problem Formulation** Building on the circuit program representation, the LS problem is defined as: *Given a truth table $\mathcal{T}$, CircuitEvo aims to generate a circuit program $\mathcal{P}$ that exactly satisfies the given functionality while minimizing the number of structure lines (circuit nodes).* To achieve this goal, CircuitEvo incorporates an evolutionary program generator and a structure-aware function optimizer into an iterative framework. Specifically, it maintains a population of $N$ functionally correct programs, denoted by $\mathcal{P} = \{P_1, \ldots, P_N\}$. In each iteration, the structure generator generates a diverse set of candidate programs, which are subsequently optimized to ensure functional correctness. The top-performing optimized programs are then added to the population for the next iteration. The overall procedure of CircuitEvo is outlined in Algorithm 1, and the core components of the circuit generator and optimizer are discussed in detail in the following sections.

## 4.2 EVOLUTIONARY PROGRAM GENERATION

Recently, the generative capabilities of large language models (LLMs) have advanced significantly, enabling their application in various complex tasks. Therefore, CircuitEvo utilizes LLMs as the program generator to explore the space of circuit programs. However, we observe that standalone LLMs with zero-shot prompting often struggle to generate high-quality programs. To address this challenge, CircuitEvo integrates LLMs within an evolutionary framework, leveraging domain-specific evolutionary prompting strategies and high-quality circuit examples to effectively guide the exploration of LLMs. The evolutionary generation process is as follows:

**Initialization** To guide the LLM in effectively exploring the circuit search space, we first construct an initial population of circuit programs that are diverse and accurate. To this end, CircuitEvo adopts a circuit decomposition strategy inspired by the Shannon decomposition theorem (Gdanskiy et al., 2020). Specifically, given the original truth table $\mathcal{T}$ with $n$ input variables, we select a decomposition variable $X_i$ and partition $\mathcal{T}$ into two sub-tables, $\mathcal{T}_1$ and $\mathcal{T}_2$, corresponding to $X_i = 1$ and $X_i = 0$, respectively:

$$\mathcal{T}(X_1, X_2, \ldots, X_n) = X_i \cdot \mathcal{T}_1(X_i = 1) + X_i' \cdot \mathcal{T}_2(X_i = 0). \tag{1}$$

We then use heuristic methods in existing LS tools to synthesize two sub-programs $\mathcal{P}_1$ and $\mathcal{P}_2$ from $\mathcal{T}_1$ and $\mathcal{T}_2$, and combine them into a functionally correct program $\mathcal{P}$ based on Equation 1. By varying the choice of decomposition variable $X_i$, this approach yields a diverse set of accurate initial solutions. More implementation details of our initialization process are provided in Appendix E.1.

---

**Algorithm 1:** Pseudo Code for CircuitEvo

**Input** : LLM $\pi_\theta$, Truth table $\mathcal{T}$, Population size $N$, $I$ iterations, $M$ strategies

**Output:** A compact and accurate circuit program

$\mathcal{P} \leftarrow \text{InitPop}(N)$
**for** $i \leftarrow 1$ **to** $I$ **do**
    **for** $m \leftarrow 1$ **to** $M$ **do**
        **for** $j \leftarrow 1$ **to** $N$ **do**
            parents $\leftarrow \text{Sample}(\mathcal{P})$
            $P_g \leftarrow \text{LLMGenerator}(\text{parents}, m)$
            $P_k \leftarrow \text{LocalSearch}(P_g)$
            $P_{opt} \leftarrow \text{FunctionOptimizer}(P_k, \mathcal{T})$
            $s_{opt} \leftarrow \text{Fitness}(P_{opt})$
            $\mathcal{P} \leftarrow \mathcal{P} \cup \{(P_{opt}, s_{opt})\}$
        **end**
    **end**
    $\mathcal{P} \leftarrow \text{GetBest}(\mathcal{P}, N)$
**end**
**return** $\text{GetBest}(\mathcal{P}, 1)$

---

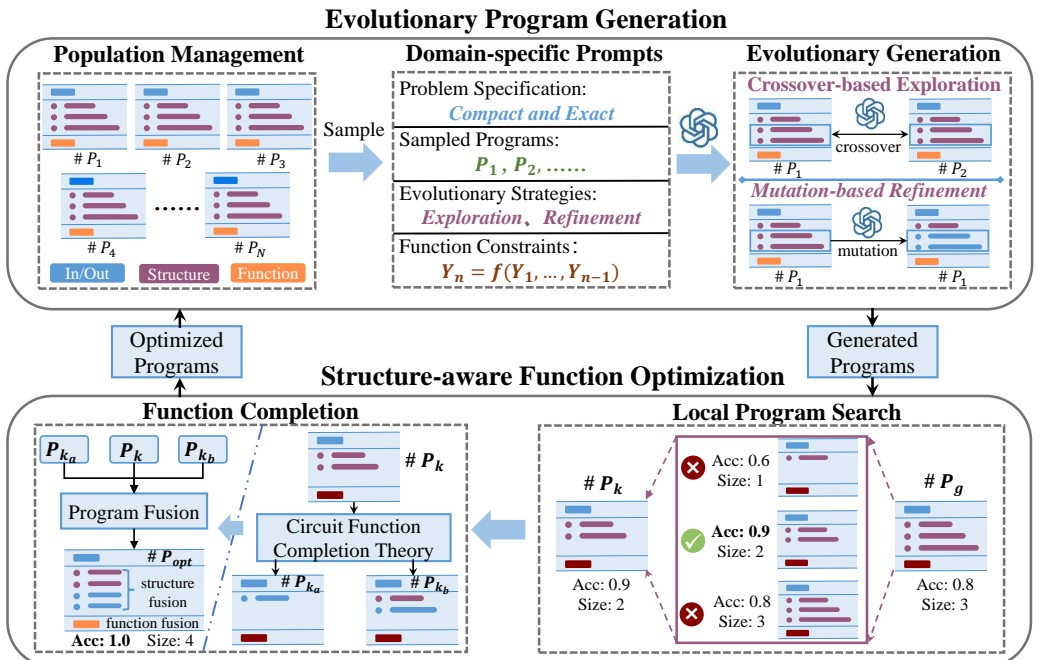

Figure 2: Overview of the CircuitEvo framework. The CircuitEvo comprises an evolutionary program generator and a structure-aware function optimizer for function completion.

**Generation of Programs** At each generation step, we sample a batch of $N$ programs $\{P_i\}_{i=1}^N$ from the LLM $\pi_\theta$ based on a carefully designed prompt $\tau_i$, i.e., $P_i \sim \pi_\theta(\tau_i)$. To fully leverage the LLM's generative capabilities for creative circuit programs generation, we construct the domain-specific evolutionary prompt as follows (see Figure 11): **Problem Specification** A detailed description of the circuit generation problem, including the program format and compactness objective. **Few-shot Learning** To stimulate the generative potential of LLMs, we adopt a few-shot learning strategy by sampling several representative circuit programs from the current population as in-context examples. To balance quality and diversity, programs are sampled with probability $\text{prob}(P_i) = \frac{1}{\text{rank}(P_i)+1+N}$, where $\text{rank}(P_i)$ indicates the fitness-based rank of $P_i$ within the population. These examples offer structural and functional priors that guide the LLM toward generating comprehensive and diverse candidate programs. **Evolutionary Prompt Strategies** To effectively leverage the sampled programs, we design four evolutionary prompt strategies to guide the LLM's generation process. These strategies are grouped into two categories: Exploration (E1, E2) and Refinement (R1, R2). The exploration strategies aim to broaden the structural search space by performing crossover-inspired transformations between pairs of parent programs. In contrast, the refinement strategies focus on improving individual programs by altering Boolean operators and variable configurations. **Domain-Specific Function Constraints** Since the circuit search space grows exponentially with the number of inputs, it is challenging to generate accurate circuit programs. To mitigate this, we incorporate domain-specific knowledge to improve generation accuracy. Specifically, we observe that the outputs of multi-output circuits often exhibit simple yet meaningful logical relationships, typically expressed as $\mathbf{Y}_m = f_m(\mathbf{Y}_1, \ldots, \mathbf{Y}_{m-1})$. For instance, in a two-output circuit, the outputs $\mathbf{Y}_1$ and $\mathbf{Y}_2$ may satisfy a relation such as $\mathbf{Y}_2 = \neg\mathbf{Y}_1$, where $\neg$ denotes the logical NOT operator. By pre-identifying inter-output dependencies and embedding them as function constraints in the prompt, the generator significantly reduces the search space. Detailed prompts are provided in Appendix E.1.

**Fitness and Population Management** In each generation, we apply all four prompt strategies $N$ times, resulting in a total of $4N$ newly generated circuit programs. These candidates are then optimized (see Section 4.3), and the feasible ones are incorporated into the current population. From this updated population, the top $N$ individuals——ranked according to their fitness scores——are selected to form the population for the next generation. Since all circuit programs in the population are functionally correct, we adopt circuit compactness (i.e., size) as the fitness metric.

### 4.3 STRUCTURE-AWARE FUNCTION OPTIMIZATION

Although CircuitEvo can continuously generate new circuit programs, the stochastic nature of evolutionary generation and the limited graph-level reasoning capabilities of large language models (LLMs) make it difficult to ensure their functional correctness. To address this problem, we present the following remark, which provides a theoretical foundation for augmenting generated programs with auxiliary substructures to ensure functional accuracy:

**Remark 1** (Circuit Functionality Completion). *Let $\mathcal{F} = \{\mathcal{F}^i\}_{i=1}^m$ denote the set of target Boolean functions defined by a truth table $\mathcal{T}$ with $m$ output bits. Let $\mathcal{F}_g = \{\mathcal{F}_g^i\}_{i=1}^m$ represent the functionality of the generated circuit programs $P_g$. Then, for each $i \in \{1, \ldots, m\}$, there exist Boolean functions $\mathcal{F}_{g_a}^i$ and $\mathcal{F}_{g_b}^i$ such that:*

$$\mathcal{F}^i = (\mathcal{F}_g^i + \mathcal{F}_{g_a}^i) * \mathcal{F}_{g_b}^i, \tag{2}$$

*where $+$ and $*$ denote the logical OR and AND operations, respectively.*

The detailed proofs are in Appendix A. Based on Remark 1, we can easily get and translate the auxiliary Boolean functions $\mathcal{F}_{g_a} = \{\mathcal{F}_{g_a}^i\}_{i=1}^m$ and $\mathcal{F}_{g_b} = \{\mathcal{F}_{g_b}^i\}_{i=1}^m$ into their corresponding truth tables $\mathcal{T}_a$ and $\mathcal{T}_b$. Subsequently, traditional heuristics are applied to generate accurate circuit programs $P_a$ and $P_b$ from these truth tables. These programs are then integrated into the initially generated program $P_g$ to complete its functionality. The program fusion process involves both structural and functional aspects. Structurally, the programs are merged with common subgraphs eliminated through structural hashing. Functionally, the completed circuit's functionality is defined by Equation 2.

**Local Program Search** Although the completion process guarantees functional correctness, it may introduce redundant logic that compromises circuit compactness. To mitigate this, we propose a local program searcher that identifies a key substructure $P_k$ within the initial program $P_g$ to guide the completion process. This approach is motivated by the observation that initial programs with higher functional accuracy and smaller sizes tend to yield more compact results after function completion, as illustrated in Figure 7. Specifically, given a circuit program comprising $n_p$ nodes, the Local Program Searcher generates $n_p$ candidate subprograms, where the $i$-th candidate consists of the first $i$ nodes of the original program. Each candidate represents a local subgraph that is structurally more compact than the full program. To identify the most promising candidate, a greedy selection strategy is applied to choose the subprogram with the highest accuracy. This localized refinement effectively eliminates redundant components, yielding a more compact and functionally accurate initial program for the function optimization. Finally, after identifying the key substructure $P_k$, traditional heuristics are employed to generate the corresponding circuit programs $\mathcal{P}_{k_a}$ and $\mathcal{P}_{k_b}$ for the final functionally optimized program $P_{opt}$. Due to limited space, we provide more implementation details about the program search and function completion process in Appendix E.2.

## 5 EXPERIMENTS

In this section, we conduct extensive experiments to evaluate CircuitEvo, which have four main parts: **Experiment 1.** To demonstrate the superior performance of our CircuitEvo in generating accurate and compact circuits. **Experiment 2.** To demonstrate that our method explores the circuit design space more efficiently than other learning-based approaches. **Experiment 3.** Perform carefully designed ablation studies to provide further insights into CircuitEvo. **Experiment 4.** Perform visualization and analysis of the inherent compact structures discovered by our method compared to the baselines.

**Benchmarks** We evaluate our method on four widely used benchmarks: Arithmetic (Alanminko, 2024), Random (Alanminko, 2024), LogicNets (He et al., 2021), and Espresso (Boucher & King, 2010). Specifically, we select 16 diverse benchmark circuits for evaluation, including very challenging cases with up to 16 inputs and 69 outputs, which corresponds to a search space of $2^{69 \times 2^{16}}$ possible solutions. Moreover, the size of the selected circuits, depending on the number of nodes synthesized by the traditional heuristic method SOP, ranges widely from 39 to 3302. Due to limited space, please refer to Appendix C.1 for more details about the benchmarks.

**Experimental Setup** Throughout all experiments, we use two open-source models—Deepseek-V3 (Liu et al., 2024a) and Qwen2.5-7B-Instruct, and a proprietary model—GPT-3.5-turbo as the backbone large language models. Although more powerful proprietary models such as GPT-4o could be used as the base model, we adopt GPT-3.5-turbo to balance generation performance with resource efficiency (see Table 6 in the Appendix). Moreover, we employ ABC (Brayton et al., 2010) as the

backend LS tool. ABC is a state-of-the-art open-source LS tool and is widely used in research of machine learning for LS (Pasandi et al., 2023; Wang et al., 2024b). All experiments are conducted on a single machine that contains 32 Intel XeonR E5-2667 v4 CPUs and 8 NVIDIA GeForce RTX 3090 GPUs. Additional implementation details can be found in Appendix C.

Table 1: The results demonstrate that CircuitEvo can generate accurate circuits for all benchmarks.

| Benchmark | Boolformer | SPL | DSR | ICSR | DNAS | CircuitEvo (Ours) | | |
|---|---|---|---|---|---|---|---|---|
| Circuit | Acc(%) ↑ | Acc(%) ↑ | Acc(%) ↑ | Acc(%) ↑ | Acc(%) ↑ | Qwen2.5-7B | Deepseek-V3 | GPT-3.5-turbo |
| Arithmetic | 89.2 | 92.0 | 89.1 | 73.9 | 100.0 | 100.0 | 100.0 | 100.0 |
| LogicNets | 97.9 | 97.9 | 96.8 | 88.1 | 99.9 | 100.0 | 100.0 | 100.0 |
| Espresso | 92.7 | 90.3 | 87.4 | 87.2 | 99.9 | 100.0 | 100.0 | 100.0 |
| Random | 87.7 | 89.0 | 87.4 | 83.7 | 100.0 | 100.0 | 100.0 | 100.0 |
| **Average** | 91.1 | 91.5 | 89.5 | 83.5 | 99.9 | **100.0** | **100.0** | **100.0** |

Table 2: The results demonstrate that CircuitEvo significantly outperforms all competitive baselines in terms of the initial size, with an average improvement of 6.74% over state-of-the-art methods.

| Benchmark | | | Boolformer | SPL | DSR | ICSR | SOP | BDD | DNAS | CircuitEvo (Ours) | | | |
|---|---|---|---|---|---|---|---|---|---|---|---|---|---|
| Circuit | PI | PO | Init Node ↓ | Init Node ↓ | Init Node ↓ | Init Node ↓ | Init Node ↓ | Init Node ↓ | Init Node ↓ | Qwen2.5-7B | Deepseek-V3 | GPT-3.5-turbo | Impr.(%) ↑ |
| Arithmetic1 | 10 | 5 | 185 | 163 | 87 | 222 | 77 | 77 | 81 | 77 | 78 | **74** | 3.90 |
| Arithmetic2 | 16 | 6 | 372 | 280 | 153 | 369 | 105 | 112 | 137 | 105 | 109 | **102** | 2.86 |
| Arithmetic3 | 16 | 7 | 1608 | 1389 | 1288 | 1512 | 1424 | 1490 | 1390 | 1229 | **1194** | 1206 | 7.30 |
| LogicNets1 | 13 | 6 | 198 | 159 | 149 | 339 | 148 | **135** | 183 | 149 | 137 | 139 | -1.48 |
| LogicNets2 | 13 | 6 | 95 | 99 | 95 | 127 | 102 | 115 | 114 | **74** | 74 | 74 | 22.11 |
| LogicNets3 | 15 | 7 | 273 | 268 | **199** | 324 | 214 | 242 | 334 | 212 | 211 | 211 | -6.03 |
| Espresso1 | 11 | 3 | 168 | 140 | 142 | 77 | 85 | 85 | 85 | **71** | 71 | 71 | 7.79 |
| Espresso2 | 11 | 9 | 287 | 309 | 244 | 396 | 175 | 173 | 140 | 133 | **128** | 128 | 8.57 |
| Espresso3 | 12 | 17 | 3874 | 3875 | 3429 | 3968 | 3302 | 3371 | 3220 | 3205 | **3184** | 3184 | 1.12 |
| Espresso4 | 16 | 69 | 911 | 2068 | 2166 | 2222 | 920 | 944 | 1157 | 863 | **848** | 848 | 6.92 |
| Random1 | 7 | 2 | 53 | 47 | 52 | 64 | 42 | 42 | **41** | 42 | 42 | 41 | 0.00 |
| Random2 | 7 | 2 | 41 | 51 | 44 | 59 | 39 | 37 | 44 | **36** | 36 | **36** | 2.70 |
| Random3 | 10 | 3 | 112 | 94 | 125 | 149 | 124 | 110 | 75 | 68 | **60** | 60 | 20.00 |
| Random4 | 10 | 3 | 153 | 173 | 153 | 169 | 154 | 156 | 152 | 140 | 147 | **138** | 9.21 |
| Random5 | 12 | 3 | 257 | 335 | 209 | 262 | 190 | 130 | 117 | **105** | 105 | 105 | 10.26 |
| Random6 | 15 | 4 | 1987 | 1808 | 1622 | 1529 | 1529 | 1514 | 1256 | 1163 | **1098** | 1110 | 12.58 |
| **Average** | | | 660.88 | 703.63 | 634.81 | 736.75 | 539.38 | 545.81 | 532.88 | **479.50** | 470.13 | 470.44 | 6.74 |

**Evaluation Metrics** In our experiments, we use the pre-mapping performance (**accuracy** and **size**) and post-mapping performance (**area** and **delay**) as the evaluation metrics to assess the accuracy and compactness of the generated circuits, respectively. (1) Accuracy is defined as the proportion of correctly predicted output bits to the total number of output bits. Specifically, it is computed as $\frac{1}{2^n \cdot m} \sum_{i=1}^{2^n} \sum_{j=1}^{m} \mathbf{1}[\hat{y}_{i,j} = y_{i,j}]$, where $n$ is the input number, $m$ is the output number, $\hat{y}_{i,j}$ is the predicted output bit, $y_{i,j}$ is the ground truth, and $\mathbf{1}[\cdot]$ denotes the indicator function. An accuracy of 100% indicates that the generated circuit fully satisfies the truth table; otherwise, the circuit does not match the truth table exactly. (2) Size is defined as the node numbers of the generated circuits. During the LS stage, reducing circuit size significantly improves the final design's PPA (Power, Performance, and Area) (Rai et al., 2021; Alanminko, 2024; Li et al., 2025). Therefore, we use the circuit size to evaluate the quality of the generated circuits. (3) Area is defined as the total physical silicon area occupied by the mapped circuit, typically measured in square micrometers, which reflects the hardware resource consumption. Delay is defined as the longest signal propagation time from any primary input to any primary output in the mapped circuit, which reflects the circuit's maximum operating speed. Smaller area and shorter delay generally indicate better hardware efficiency, so these metrics are used to assess the practical performance of the generated circuits on a target technology.

**Baselines** Our baselines fall into two categories: traditional methods and learning-based methods. For traditional methods, we employ two widely used heuristics: **SOP** (Nabulsi et al., 2017) and **BDD** (Lai et al., 1993). For learning-based methods, we utilize the state-of-the-art (SOTA) machine learning technique **DNAS** (Wang et al., 2024b; Chu et al.). Moreover, we employ four Boolean function learning methods, including a pre-trained method, two search-based techniques, and an LLM-based method. Specifically: (1) **Boolformer** (d'Ascoli et al., 2023) is a pre-trained model designed for Boolean function learning in circuit generation; (2) **SPL** (Sun et al., 2022) is a SOTA Monte Carlo Tree Search-based function learning method; (3) **DSR** (Petersen et al., 2019) is a search-based function learning method with reinforcement learning; (4) **ICSR** (Merler et al., 2024) is an LLM-based function learning approach. More details about the baselines are presented in Appendix D.

**Experiment 1. Generation performance analysis** In this subsection, we compare CircuitEvo with seven competitive baselines in terms of pre-mapping metrics (accuracy and size), and post-mapping metrics (area and delay). The evaluation is conducted on four widely used circuit benchmarks. Results in Table 1 show that our CircuitEvo outperforms all learning-based methods in terms of

Table 3: The results demonstrate that CircuitEvo significantly outperforms baselines in terms of the circuit area and delay after technology mapping on four benchmarks.

| Benchmark | Boolformer | SPL | DSR | ICSR | SOP | BDD | DNAS | CircuitEvo |
|---|---|---|---|---|---|---|---|---|
| | Area ↓ | Area ↓ | Area ↓ | Area ↓ | Area ↓ | Area ↓ | Area ↓ | Area ↓ |
| Arithmetic | 1232.67 | 1037.67 | 878.67 | 1131.67 | 915.33 | 940.00 | 928.00 | **789.67** |
| Espresso | 2266.00 | 2410.00 | 2553.50 | 2504.50 | 1894.25 | 1912.75 | 2035.50 | **1818.25** |
| LogicNets | 323.00 | 303.67 | 259.67 | 364.33 | 274.00 | 287.33 | 403.67 | **251.33** |
| Random | 749.67 | 717.67 | 642.50 | 623.17 | 590.17 | 572.33 | 511.67 | **431.33** |
| | Delay ↓ | Delay ↓ | Delay ↓ | Delay ↓ | Delay ↓ | Delay ↓ | Delay ↓ | Delay ↓ |
| Arithmetic | 17.73 | 16.57 | 15.23 | 15.30 | 15.60 | 15.90 | 14.07 | **13.67** |
| Espresso | 14.55 | 15.18 | 15.98 | 16.25 | 15.45 | 15.48 | 14.58 | **14.15** |
| LogicNets | 12.27 | 12.70 | 12.43 | 12.43 | 12.30 | 12.37 | 13.10 | **12.23** |
| Random | 13.22 | 13.27 | 12.63 | 12.62 | 13.17 | 12.93 | 11.67 | **10.80** |

Table 4: The ablation results show that each component plays an important role in our CircuitEvo.

| Benchmark | CircuitEvo (Ours) | | w/o LLM | | w/o Program | | w/o Evolution | | w/o Local | | w/o Completion | |
|---|---|---|---|---|---|---|---|---|---|---|---|---|
| Circuit | Acc(%) ↑ | Init Node ↓ | Acc(%) ↑ | Init Node ↓ | Acc(%) ↑ | Init Node ↓ | Acc(%) ↑ | Init Node ↓ | Acc(%) ↑ | Init Node ↓ | Acc(%) ↑ | Init Node ↓ |
| Arithmetic1 | 100.0 | **74** | 100.0 | 78 | 100.0 | 81 | 100.0 | 78 | 100.0 | 79 | 100.0 | 74 |
| Arithmetic3 | 100.0 | 1206 | 100.0 | 1229 | 100.0 | 1229 | 100.0 | 1233 | 100.0 | **1178** | 67.8 | 27 |
| Espresso3 | 100.0 | **3184** | 100.0 | 3220 | 100.0 | 3377 | 100.0 | 3184 | 100.0 | 3337 | 82.6 | 200 |
| Espresso4 | 100.0 | **848** | 100.0 | 940 | 100.0 | 940 | 100.0 | 911 | 100.0 | 911 | 86.4 | 9 |
| LogicNets1 | 100.0 | **139** | 100.0 | 140 | 100.0 | 148 | 100.0 | 150 | 100.0 | 140 | 70.1 | 2 |
| LogicNets2 | 100.0 | **74** | 100.0 | 90 | 100.0 | 74 | 100.0 | 75 | 100.0 | 75 | 60.6 | 3 |
| Random4 | 100.0 | **138** | 100.0 | 147 | 100.0 | 146 | 100.0 | 151 | 100.0 | 144 | 84.5 | 20 |
| Random5 | 100.0 | **105** | 100.0 | 110 | 100.0 | 106 | 100.0 | 106 | 100.0 | 105 | 83.3 | 49 |

accuracy. Specifically, CircuitEvo achieves an average improvement of 16.5% compared to the LLM-based baseline, ICSR. Additionally, all circuits generated by CircuitEvo achieve 100% accuracy, demonstrating the effectiveness of our structure-aware function optimizer. To fairly compare circuit sizes, we first apply the rule-based legalization method from (Wang et al., 2024b) to ensure that circuits generated by baseline methods are functionally correct, i.e., 100% accuracy. Building upon that, the results in Table 2 show that our CircuitEvo significantly outperforms all baselines in terms of the generated circuit size. Specifically, CircuitEvo achieves an average reduction of 6.74% in size compared to the SOTA baselines. Furthermore, to demonstrate that the circuits produced by our method provide better initial solutions for optimization, we apply three optimization heuristics—*resyn2*, *dc2*, and *compress2*—to the generated circuits. Results in Table 8 show that our method achieves an average reduction of 11.09% in the optimized circuit size compared to the SOTA baselines under the *resyn2*, indicating the superiority of our method.

Moreover, to assess the practical efficiency of the generated circuits after technology mapping, we evaluate their post-mapping performance in terms of area and delay. Specifically, we use ABC's `map` operator to map each circuit onto `mcnc.genlib`, a widely used standard cell library with 23 different gates in logic synthesis, and measure the resulting area and delay. As shown in Table 3, CircuitEvo consistently achieves smaller area and shorter delay than all baselines across the four benchmarks, indicating that our method not only generates functionally correct and compact circuits but also produces designs that translate into more efficient hardware implementations. Overall, these comparative results demonstrate that CircuitEvo surpasses existing methods in generating accurate and compact circuits. More comparative results are provided in Appendix F.1.

**Experiment 2. Efficiency analysis** In this subsection, we compare the search efficiency of our method with three competitive search-based baselines across circuits from four benchmarks. For a fair comparison, we adopt convergence time as the metric for evaluating efficiency. The results in Table 5 demonstrate that CircuitEvo consistently outperforms all baselines in terms of search efficiency, demonstrating its ability to effectively navigate the search space by leveraging the powerful

Table 5: The efficiency comparative results.

| Benchmark | CircuitEvo | DSR | SPL | DNAS |
|---|---|---|---|---|
| Circuit | Time(h) ↓ | Time(h) ↓ | Time(h) ↓ | Time(h) ↓ |
| Arithmetic1 | **1.21** | 2.02 | 3.88 | 7.69 |
| LogicNets1 | **1.34** | 3.73 | 3.99 | 32.16 |
| Espresso1 | **2.04** | 4.01 | 3.54 | 18.03 |
| Random5 | **1.32** | 2.38 | 4.25 | 9.75 |
| **Average** | **1.48** | 3.91 | 3.03 | 16.91 |

search capability of LLMs. Overall, the results demonstrate the superior efficacy of our CircuitEvo.

**Experiment 3. Ablation Study** In this subsection, we conduct an ablation study on eight circuits using GPT-3.5-turbo as the LLM backbone to understand the individual contribution of each component

of our CircuitEvo (see Table 4). The *'w/o LLM'* variant, which removes the use of LLM and instead employs a rule-based genetic algorithm—linear genetic program (Oltean & Grosan, 2003)—to evolve the circuits, led to a considerable size expansion. This highlights the importance of incorporating the strong generation ability and prior knowledge of LLM in our CircuitEvo. The *'w/o Program'* variant, which formulates the circuit as a Boolean function rather than a circuit program, led to a substantial increase in size. This highlights that our proposed graph-structured program captures more structural and functional information than other textual representations. The *'w/o Evolution'* variant, which doesn't use the exploration and refinement evolutionary strategies for circuit generation, also exhibits a negative impact on size. This demonstrates the effectiveness of incorporating LLM in an evolutionary framework. The *'w/o Local'* variant, which completes the originally generated circuits without applying local program search, results in a significant increase in size. This demonstrates the necessity of local search in eliminating redundant structures. The *'w/o Completion'* variant, which removes the function completion approach and directly uses the size of the LLM's generated circuits as the fitness metric, results in a substantial drop in accuracy. This underscores the critical role of our proposed function optimizer in ensuring the functional correctness of the final generated circuits.

**Experiment 4. Visualization of compact structures** In this subsection, we aim to provide insights into the compact structural patterns captured by our method. By visualizing the generated circuit structures, we observe that the compactness achieved by our approach primarily manifests in two aspects: *increased logic sharing* and the emergence of *triangular circuit structures*. Due to limited space, we provide more visualization results in Appendix F.3.

**Logic sharing** is defined as the logic nodes or subexpressions that are shared by multiple logical components. For example, $X_2X_3$ is a logic sharing that appears twice in the expression $X_1X_2X_3 * (X_2X_3 + X_4)$ and can be shared as a node in the final circuit. By analyzing the generated circuits, we found that *increased logic sharing always leads to reduced circuit size*. Specifically, the results in Figure 3 show that our method, which generates more compact circuits, achieves a significantly larger number of logic sharing than all Boolean function learning baselines on four benchmarks. This observation underscores the importance of logic sharing in reducing circuit size and suggests that future methods should explicitly promote its exploitation in circuit generation.

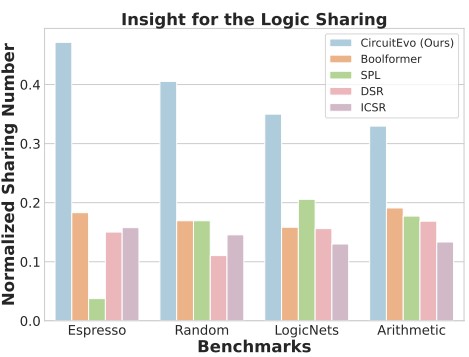

Figure 3: Logic Sharing significantly matters with the circuit compactness.

**Triangle structure** refers to a circuit where the number of lower-level nodes consistently exceeds that of higher-level nodes. The current logic network learning methods necessitate the definition of an initial circuit structure to guide the search (Belcak & Wattenhofer, 2022b; Wang et al., 2024b). Nevertheless, a principled methodology for designing these initial structures remains lacking. Through the analysis of generated circuits, we have observed that *compact circuits consistently exhibit a triangle structure*. Specifically, Figure 4 shows that the number of nodes decreases as the circuit level increases on five benchmark circuits, confirming the presence of a triangular structure. Therefore, it is suggested that triangular networks should be prioritized in the initial network design, which can lead to better circuit generation.

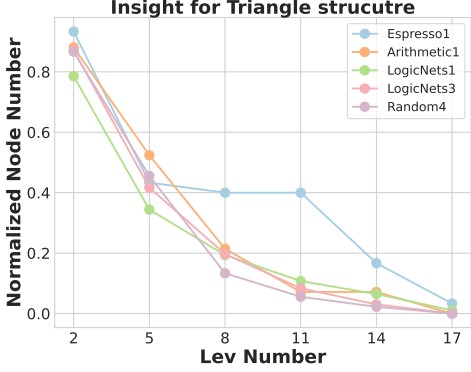

Figure 4: Triangle Structures play a significant role in circuit compactness.

## 6 CONCLUSION

Logic synthesis aims to generate a compact and accurate logic circuit, which plays an important role in chip design. However, existing LS methods struggle to balance circuit structure compactness and functional accuracy, often leading to suboptimal generation. To address this problem, we propose a

novel Circuit Program Evolution framework, namely CircuitEvo. This framework models the circuit graph as a structured program and integrates an LLM-based program generator and structure-aware function optimizer to explore the circuit program space. An appealing feature of this framework lies in its ability to enable iterative improvement of circuit compactness while preserving functional accuracy. Experiments on several benchmarks demonstrate that CircuitEvo can efficiently generate accurate circuits and significantly outperforms the state-of-the-art methods in terms of the circuit size.

## ETHICS STATEMENT

This research does not involve any personally identifiable information. All datasets used are publicly available and widely adopted in the community, and we have verified that their licenses permit research use. In accordance with the ICLR Code of Ethics (`https://iclr.cc/public/CodeOfEthics`), we ensure that our work adheres to principles of fairness, transparency, and responsible AI research. We also disclose that LLMs were used for text polishing, while all conceptual contributions and validation remain the responsibility of the authors in Appendix G.

## REPRODUCIBILITY STATEMENT

We will provide open access to all source code, configuration files, and preprocessing scripts, together with detailed instructions to reproduce the main experimental results. All datasets employed are publicly available, and we specify the exact versions and preprocessing steps. We report all hyperparameters, model versions, and API parameters in full, and we describe the computational environment (hardware type, GPU model, and software dependencies) in the supplemental material. We also include ablation studies and negative results to ensure transparency. Collectively, these resources and specifications enable reliable and faithful reproduction of our results.

## ACKNOWLEDGMENTS

This work was supported in part by National Key R&D Program of China under contract 2022ZD0119801, National Nature Science Foundations of China grants U23A20388 and 62021001. This work was supported in part by Huawei as well. We would like to thank all the anonymous reviewers for their insightful comments.

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

## A PROOF OF CIRCUIT FUNCTIONALITY COMPLETION (REMARK 1)

Before proceeding with the proof, we briefly recall the key symbols and definitions involved. Throughout this section, we use $+$, $*$, and $\neg$ to denote the logical OR, AND, and NOT operations, respectively. $\mathcal{F} = \{\mathcal{F}^i\}_{i=1}^m$ denotes the set of target Boolean functions defined by the truth table $\mathcal{T}$ with $m$ output bits. $\mathcal{F}_g = \{\mathcal{F}_g^i\}_{i=1}^m$ represent the functionality of the generated circuit programs $P_g$. Our goal is to prove that for each $i \in \{1, \ldots, m\}$, there exist Boolean functions $\mathcal{F}_{g_a}^i$ and $\mathcal{F}_{g_b}^i$ such that:

$$\mathcal{F}^i = (\mathcal{F}_g^i + \mathcal{F}_{g_a}^i) * \mathcal{F}_{g_b}^i.$$

From Boolean algebra, we know that Boolean functions have the following properties. Let $F$, $G$ and $H$ be Boolean functions, we have:

- Associativity of AND:    $F * (G * H) = (F * G) * H$
- Associativity of OR:    $F + (G + H) = (F + G) + H$
- Commutativity of AND:    $F * G = G * F$
- Commutativity of OR:    $F + G = G + F$
- Distributivity of AND over OR:    $F * (G + H) = (F * G) + (F * H)$
- Identity for AND:    $F * 1 = F$
- Identity for OR:    $F + 0 = F$
- Annihilator for AND:    $F * 0 = 0$
- Annihilator for OR:    $F + 1 = 1$
- Idempotence of AND:    $F * F = F$
- Idempotence of OR:    $F + F = F$
- Complementation of AND:    $F * (\neg F) = 0$
- Complementation of OR:    $F + (\neg F) = 1$

*Proof.* For each $i \in \{1, \ldots, m\}$, we define $\mathcal{F}_{g_a}^i$ and $\mathcal{F}_{g_b}^i$ as

$$\mathcal{F}_{g_a}^i = \mathcal{F}^i * (\neg \mathcal{F}_g^i) \quad \text{and} \quad \mathcal{F}_{g_b}^i = \mathcal{F}^i + (\neg \mathcal{F}_g^i) \tag{3}$$

respectively. Thus, according to the distributivity of AND over OR, the expression can be rewritten as

$$\begin{aligned}
(\mathcal{F}_g^i + \mathcal{F}_{g_a}^i) * \mathcal{F}_{g_b}^i &= (\mathcal{F}_g^i + (\mathcal{F}^i * (\neg \mathcal{F}_g^i))) * (\mathcal{F}^i + (\neg \mathcal{F}_g^i)) \\
&= \mathcal{F}_g^i * \mathcal{F}^i + \mathcal{F}_g^i * (\neg \mathcal{F}_g^i) + (\mathcal{F}^i * (\neg \mathcal{F}_g^i)) * \mathcal{F}^i + (\mathcal{F}^i * (\neg \mathcal{F}_g^i)) * (\neg \mathcal{F}_g^i).
\end{aligned}$$

According to the above properties, we can simplify $\mathcal{F}_g^i * (\neg \mathcal{F}_g^i)$ to $0$, $(\mathcal{F}^i * (\neg \mathcal{F}_g^i)) * \mathcal{F}^i$ to $\mathcal{F}^i * (\neg \mathcal{F}_g^i)$ and $(\mathcal{F}^i * (\neg \mathcal{F}_g^i)) * (\neg \mathcal{F}_g^i)$ to $\mathcal{F}^i * (\neg \mathcal{F}_g^i)$. Substituting this result back into the original expression, we have

$$\begin{aligned}
(\mathcal{F}_g^i + \mathcal{F}_{g_a}^i) * \mathcal{F}_{g_b}^i &= \mathcal{F}_g^i * \mathcal{F}^i + 0 + \mathcal{F}^i * (\neg \mathcal{F}_g^i) + \mathcal{F}^i * (\neg \mathcal{F}_g^i) \\
&= \mathcal{F}_g^i * \mathcal{F}^i + \mathcal{F}^i * (\neg \mathcal{F}_g^i) + \mathcal{F}^i * (\neg \mathcal{F}_g^i) \\
&= \mathcal{F}_g^i * \mathcal{F}^i + \mathcal{F}^i * (\neg \mathcal{F}_g^i) \\
&= \mathcal{F}^i * (\mathcal{F}_g^i + (\neg \mathcal{F}_g^i)) \\
&= \mathcal{F}^i * 1 \\
&= \mathcal{F}^i,
\end{aligned}$$

which completes our proof. $\square$

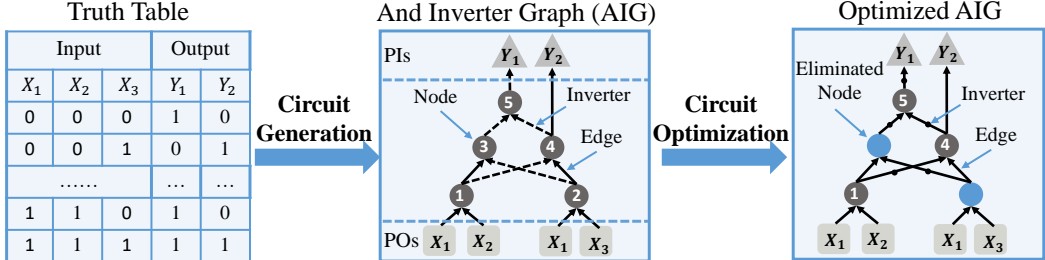

Figure 5: The illustration of two stages in LS: Circuit Generation and Circuit Optimization.

## B   MORE DETAILS FOR BACKGROUND AND RELATED WORK

**Circuit Graph Representation** In the LS stage, circuits are typically represented using And-Inverter Graphs (AIGs) (Berndt et al., 2022). An AIG is a directed acyclic graph where each node represents a Boolean function and each edge corresponds to a wire connecting these functions. AIGs include four types of nodes: constants, primary inputs (PIs), primary outputs (POs), and two-input AND gates (And2 nodes). The graph edge is either inverted or not, with a complemented edge indicating that the signal is inverted. Given a truth table with $n$ inputs and $m$ outputs, the AIG contains $n$ PIs and $m$ POs. The size of a circuit denotes the number of And2 nodes in the AIG, while the depth/level signifies the longest path from a PI to a PO.

**Symbolic Regression (SR) for Learning Boolean Functions** Symbolic Regression (SR) aims to discover a mathematical expression $f$ that best fits a given dataset $\mathcal{D}$. To achieve this, many existing SR approaches represent mathematical expressions as algebraic expression trees, where internal nodes are operators (e.g., $+, \times, \sin$) and terminal nodes are input variables or constants. We denote by $\tau = [\tau_1, \ldots, \tau_n]$ the pre-order traversal of such an expression tree. Notably, there exists a one-to-one correspondence between an expression tree and its pre-order traversal. Each $\tau_i$ is an operator, input variable, or constant selected from a predefined library of tokens, such as $[+, -, \times, \div, \sin, \cos, \exp, \log, x]$. In Logic Synthesis (LS), Boolean functions can be learned from truth tables using SR methods. To adapt existing SR techniques for Boolean function learning, we redefine the operator set as $\text{and}, \text{or}, \text{not}$, and the token library at each step becomes $[\text{and}, \text{or}, \text{not}, x_1, \ldots, x_n]$. Once these functions are learned, they can be directly mapped into logic circuits for further synthesis or optimization.

**LLM with Evolutionary methods** Evolutionary Computation (EC), inspired by the fundamental mechanisms of natural selection and biological evolution, has emerged as a powerful and flexible optimization paradigm (Katoch et al., 2021; Gen & Lin, 2023). Recent advancements demonstrate the significant potential of incorporating EC into the prompt engineering of large language models (LLMs), leading to notable performance improvements across diverse tasks, including code generation, natural language understanding, and the automated design of heuristics (Hemberg et al., 2024; Romera-Paredes et al., 2024; Liu et al., 2024b; Yao et al., 2024; Morris et al., 2024). By synergizing EC with prompt engineering, it becomes possible to effectively navigate and exploit the vast and complex search space inherent to LLM-driven tasks. This integration not only facilitates more efficient exploration and exploitation strategies but also enhances the model's ability to uncover higher-quality solutions that may be difficult to identify through conventional optimization approaches.

## C   IMPLEMENTATION DETAILS OF EXPERIMENTAL SETUP

### C.1   DETAILS ON THE BENCHMARKS

We provide the detailed information of the circuits used in our paper as follows.

- **Arithmetic 1, 2, 3**: Arithmetic functions with permuted inputs / dropped outputs.
- **Espresso 1, 2, 3, 4**: Selected Espresso benchmarks with permuted inputs.
- **LogicNets 1, 2, 3**: Three-output neurons from the LogicNets (He et al., 2021) project.
- **Random 1, 2, 5, 6**: Random and modified random functions.

- **Random 3, 4**: Known random-looking functions.

## C.2 HYPERPARAMETERS

In our experiments, we implemented the CircuitEvo framework using GPT-3.5-TURBO and DEEPSEEK-V3 as the base LLM generators ($\pi_\theta$), with a population size of $N = 10$ and $I = 10$ iterations to balance exploration and computational efficiency. The evolutionary process employed $M = 4$ distinct strategies: two crossover-based exploration methods (E1, E2) for generating novel circuit designs by combining parent solutions, and two mutation-based refinement techniques (R1, R2) for fine-tuning individual circuits through local modifications. Moreover, for the learning-based baseline methods, we use the same default hyperparameters as reported in the corresponding paper.

## C.3 USE OF LS TOOLS

In our paper, we leverage the Logic Synthesis (LS) tool ABC Brayton et al. (2010) in two key aspects. First, we adopt ABC's traditional heuristic methods—Sum-of-Products (SOP) and Binary Decision Diagrams (BDD)—as baselines and functional optimizers. These heuristics are applied to the Random1 circuit using the following operator sequences:

- **SOP**: *read_truth -xf Random1.truth; collapse; sop; strash*
- **BDD**: *read_truth -xf Random1.truth; bdd; strash*

Second, during the circuit optimization stage, we employ three widely used ABC operators—resyn2, dc2, and compress2—to refine the initial synthesized circuits. The specific optimization sequences are:

- **resyn2**: `read Random1.aig; strash; resyn2`
- **dc2**: `read Random1.aig; strash; dc2;`
- **compress2**: `read Random1.aig; strash; compress2;`

Here, `.aig` denotes the file format of the generated And-Inverter Graph (AIG) circuits.

## D IMPLEMENTATION DETAILS OF THE BASELINES

Below, we provide short descriptions of the five learning-based methods and two manually designed heuristics.

- **DNAS**: DNAS (Wang et al., 2024b) is a differentiable neural architecture search method designed to synthesize circuits from input-output examples. It begins by constructing a logic network architecture and then employs gradient descent to train the network. After training, DNAS prunes unnecessary nodes and outputs the remaining structure as the circuit network. However, the search quality heavily depends on the initial network. An inappropriate initial network always hinders the learning-based methods from discovering compact structures. In logic synthesis, traditional neural architecture search methods (Wang et al., 2024b; Petersen et al., 2022) often use square structures, which introduce significant structural bias compared to the inherent circuit structure, leading to inefficient search processes.

- **Boolformer**: Boolformer (d'Ascoli et al., 2023) is a pioneering method that introduces the Transformer framework to learn compact Boolean functions for logic synthesis. This innovative approach significantly advances the field by combining the strengths of a pre-trained model and symbolic reasoning.

- **DSR**: DSR Petersen et al. (2019) is a search-based method that combines reinforcement learning with a recurrent neural network to generate highly expressive mathematical expressions. By leveraging the strengths of reinforcement learning, DSR efficiently explores the space of possible expressions, while the recurrent neural network enables the model to capture complex patterns and dependencies within the data.

- **SPL**: SPL Sun et al. (2022) is a state-of-the-art search-based method that utilizes a Monte Carlo Tree Search (MCTS) agent to systematically explore and identify optimal expression trees from measurement data. By leveraging the strategic exploration capabilities of MCTS, SPL efficiently navigates the vast search space of potential expressions, ensuring a balance between exploration and exploitation.

- **ICSR**: ICSR Merler et al. (2024) is an LLM-based symbolic regression method that utilizes large language models to search for optimal expressions. A key strength of ICSR lies in its ability to iteratively refine generated expressions using off-the-shelf optimizers. This approach effectively combines the exploratory power of LLMs with the domain-specific expertise of external tools.

- **SOP**: SOP (Nabulsi et al., 2017) is a heuristic method integrated into the widely used logic synthesis framework ABC. This method achieves precise logic synthesis by representing truth tables as sums of products.

- **BDD**: BDD (Lai et al., 1993) is another heuristic method included in the ABC logic synthesis framework. It achieves precise logic synthesis by applying the Shannon decomposition theorem and representing the truth table as a binary decision diagram (Bryant, 1992; Akers, 1978; Drechsler & Becker, 2013).

# E  IMPLEMENTATION DETAILS OF OUR CIRCUITEVO

## E.1  MORE DETAILS OF THE EVOLUTIONARY PROGRAM GENERATION FRAMEWORK

**Initialization process** In the initialization phase, we adopt a circuit decomposition strategy to generate a diverse and accurate initial population. Specifically, the original truth table $\mathcal{T}$ is decomposed into two sub-truth tables, $\mathcal{T}_1$ and $\mathcal{T}_2$, according to Equation 1. Each sub-truth table is then converted into a circuit graph using the traditional heuristic SOP Nabulsi et al. (2017), and subsequently converted into programs $P_1$ and $P_2$. These two programs are then fused into an accurate program $P$ (see Section E.2 for the fusion method). To ensure diversity in the initial population, we vary the decomposition variable and repeat the process $N$ times, resulting in $N$ candidate programs. Each sample corresponds to a different decomposition variable, enabling the generation of a diverse set of circuit programs.

**The domain-specific evolutionary prompts** We present the detailed prompts in Figure 11. Specifically, the prompts are structured into four main components: Problem Specification, which defines the task and objectives; Circuit Program Example, which offers a concrete example to guide the generation process; Evolutionary Generation Strategies, which describe the methods for generating and evolving candidate solutions; and Domain-Specific Function Constraints, which incorporate domain knowledge to effectively reduce the search space.

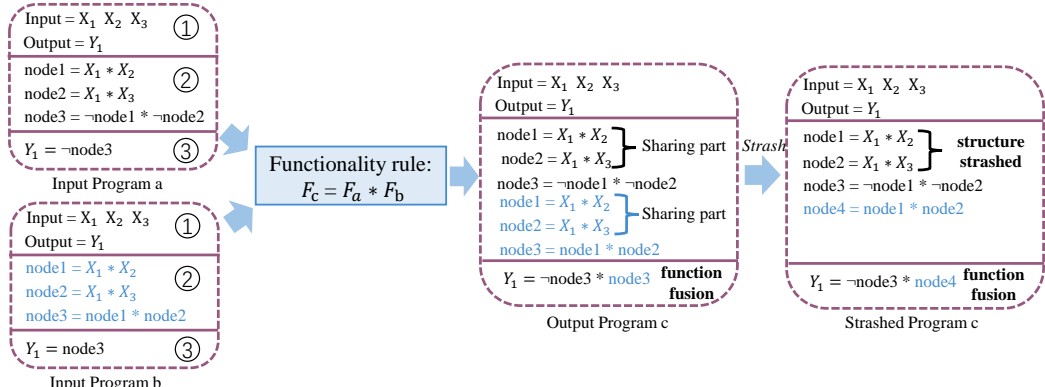

Figure 6: An illustration of the program fusion process. Two input programs are fused to get a new program based on a given functionality rule and structure hashed.

### E.2 MORE DETAILS OF THE STRUCTURE-AWARE FUNCTION OPTIMIZATION FRAMEWORK

**Program Fusion** The circuit program fusion process, as illustrated in Figure 6, can be clearly divided into two complementary dimensions: structural fusion and functional fusion. **Structurally**, the fusion begins by directly stacking the logic structures from both input programs. At this stage, all nodes from both programs are combined into a single unified representation without any simplification. This forms an intermediate circuit that contains potentially duplicated logic patterns across the two input structures. To optimize this intermediate structure, we employ the *strash* operator from existing LS tools, which performs structural hashing. This operation identifies and merges *shared subgraphs*— logic nodes that compute the same Boolean functions—thereby removing redundancy and yielding a more compact and efficient circuit. In the figure, such redundant substructures (e.g., $X_1 * X_2$, $X_1 * X_3$) are merged to form the structure of the final program. **Functionally**, the fusion proceeds by combining the output functionalities of the input programs according to a predefined rule. In the given example, the functional rule is $F_c = F_a * F_b$, indicating that the final circuit should output the logical conjunction of the functionalities of two input programs. This combination is implemented by constructing a new output expression based on the original outputs ($\neg node3$ from Program a and $node3$ from Program b). The fused functional output is thus defined as $Y_1 = \neg node3 * node4$. In summary, the fusion process simultaneously operates on the structural and functional levels: the structure is stacked and then optimized through strash, while the function is fused directly based on a given rule. This two-step design ensures both correctness and compactness in the resulting program.

**Motivations for Local Program Search** Although the circuit functionality completion theory ensures the accuracy of the circuit programs, it often leads to an increase in circuit size. Therefore, a key challenge is to identify an initial circuit program that minimizes this size overhead after function optimization. To address this issue, we propose a local program search method that explores the neighborhood of the initially generated program to discover candidates with higher accuracy and smaller size. This approach is motivated by our empirical observation, illustrated in Figure 7, that initial circuits with higher functional accuracy and more compact structure result in better circuits.

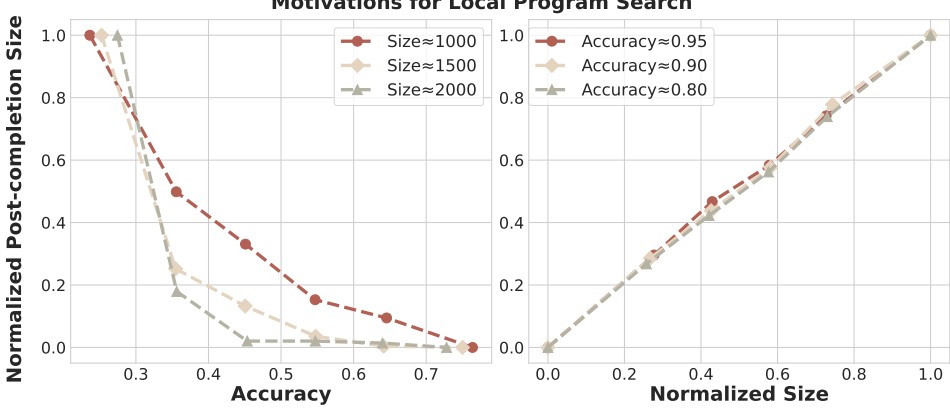

Figure 7: Motivations for Local Program Search. Experimental results show that, with fixed accuracy, larger pre-completion circuits lead to larger final circuits, while with fixed pre-completion size, higher accuracy results in smaller final circuits. These insights motivate us to generate smaller and more accurate pre-completion circuits.

## F ADDITIONAL RESULTS

### F.1 MORE RESULTS FOR COMPARATIVE EVALUATION

In this subsection, we provide more comparative results. Specifically, we evaluate our method on sixteen circuits from four widely used benchmarks. In terms of the accuracy of the generated circuit, Table 7 demonstrates that all circuits generated by CircuitEvo achieve 100% accuracy. In terms of the initial size, Table 2 shows that our CircuitEvo significantly outperforms all competitive baselines in terms of the initial size, with an average improvement of 6.74% over state-of-the-art methods. In terms of the optimized size, the results in Table 8, Table 9, and Table 10 demonstrate that our

CircuitEvo achieves an average improvement of 11.09%, 9.91%, and 10.79% in optimized circuit size compared to the state-of-the-art methods when using *resyn2*, *dc2* and *compress2* optimization operator, respectively. Overall, the comparative results demonstrate that CircuitEvo can effectively generate more accurate and compact circuits than the baseline methods.

## F.2 THE EVOLUTION TRAJECTORY RESULTS OF OUR CIRCUITEVO

In this subsection, we present the evolution trajectory results of our method on two circuits. As shown in Figure 8, CircuitEvo can iteratively improve both the average and best performance of the population over iterations, indicating the effectiveness of our method.

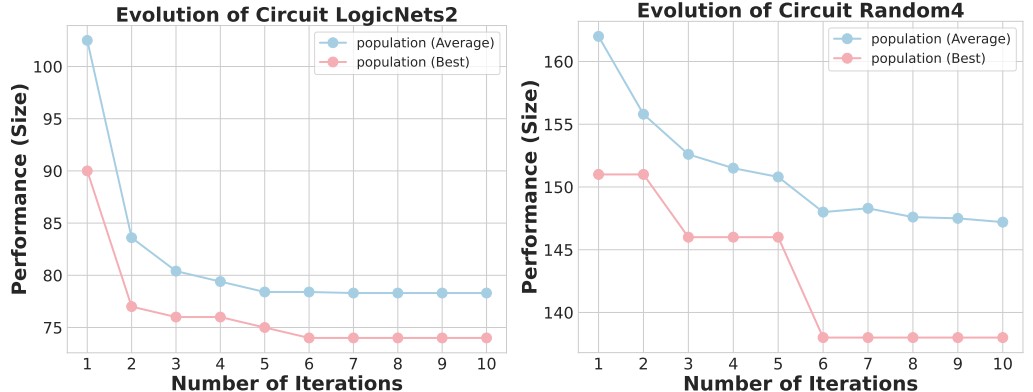

Figure 8: From left to right: (a) The evolutionary trajectory of Circuit LogicTNets2. (b) The evolutionary trajectory of Circuit Random4. The blue line indicates the average performance of the population at each iteration, while the red line represents the performance of the best individual.

## F.3 MORE VISUALIZATION RESULTS OF OUR CIRCUITEVO

In this subsection, we visualize the generated circuit programs and present additional analysis on two key factors that contribute to circuit compactness: logic sharing and triangular structures.

**Logic Sharing** As shown in Figure 9, logic sharing is defined as the logic nodes or sub-expressions that are shared by multiple logical components. The detailed results in Table 11 demonstrate that our method, which generates more compact circuits, captures a significantly larger number of logic sharing than all function learning baselines. Therefore, we can conclude that increased logic sharing corresponds to reduced circuit size. To further illustrate this, we visualize the Random3 circuit program and graph generated by CircuitEvo in Figure 10. The

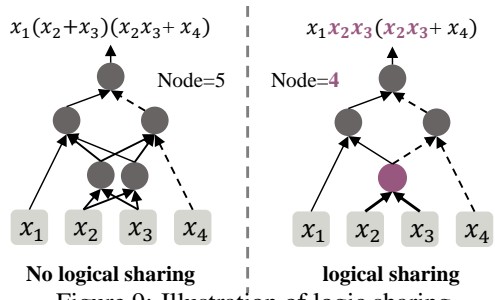

Figure 9: Illustration of logic sharing.

visualization reveals that approximately one-third of the nodes are shared across different parts of the circuit, providing evidence of the compactness enabled by logic sharing.

**Triangular Structures** In this subsection, we provide additional results to support the presence of triangular structures in our generated circuits. Specifically, the detailed results in Table 12 and Table 13 consistently demonstrate this structural pattern across all test circuits. Furthermore, visualization results shown in Figure 10 offer intuitive evidence for the existence of the triangular structure.

## G THE USE OF LARGE LANGUAGE MODEL

In accordance with ICLR 2026 policy, we disclose the use of Large Language Models (LLMs) as an assistive tool in the preparation of this manuscript. The primary application of LLMs was to aid in improving the clarity and quality of the writing.

Our process involved using an LLM to perform the following specific tasks:

- **Grammar Correction:** Identifying and correcting grammatical errors and spelling mistakes.
- **Clarity and Readability Enhancement:** Rephrasing sentences and suggesting alternative phrasings to improve the overall readability and flow of the text.
- **Conciseness:** Assisting in shortening sentences and paragraphs to make the writing more direct and concise.

The core scientific contributions, analyses, and claims presented in this paper are the work of the human authors. We have ensured that the use of LLMs in the writing process was conducted responsibly and in line with academic and ethical standards.

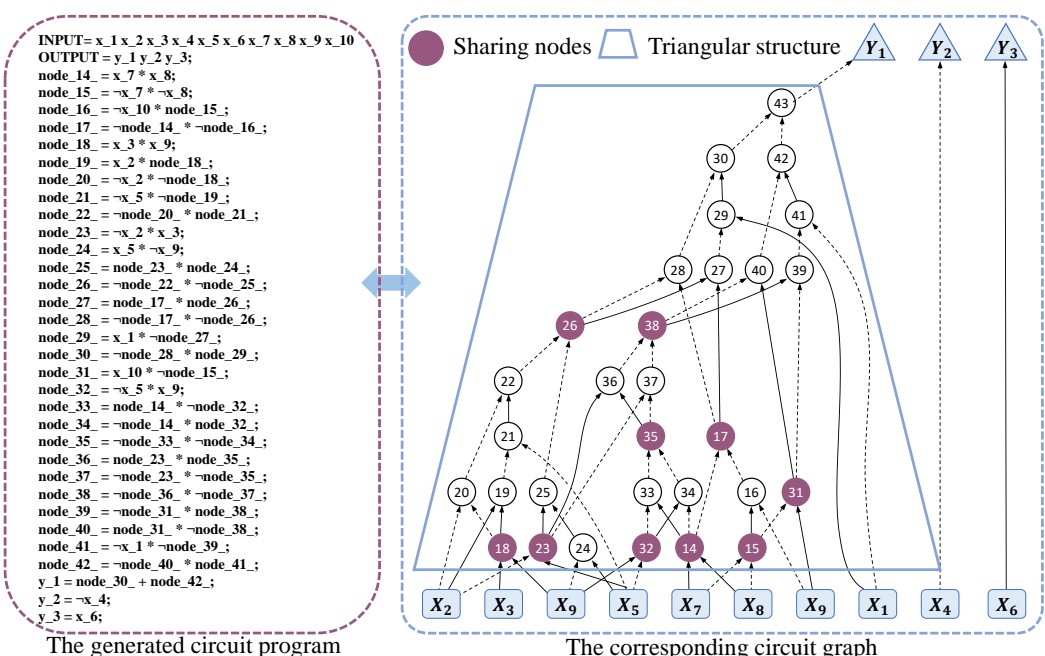

Figure 10: Visualization of the generated circuit program and its corresponding AIG graph for the Random3 circuit. The logic sharing and triangular structure are visualized.

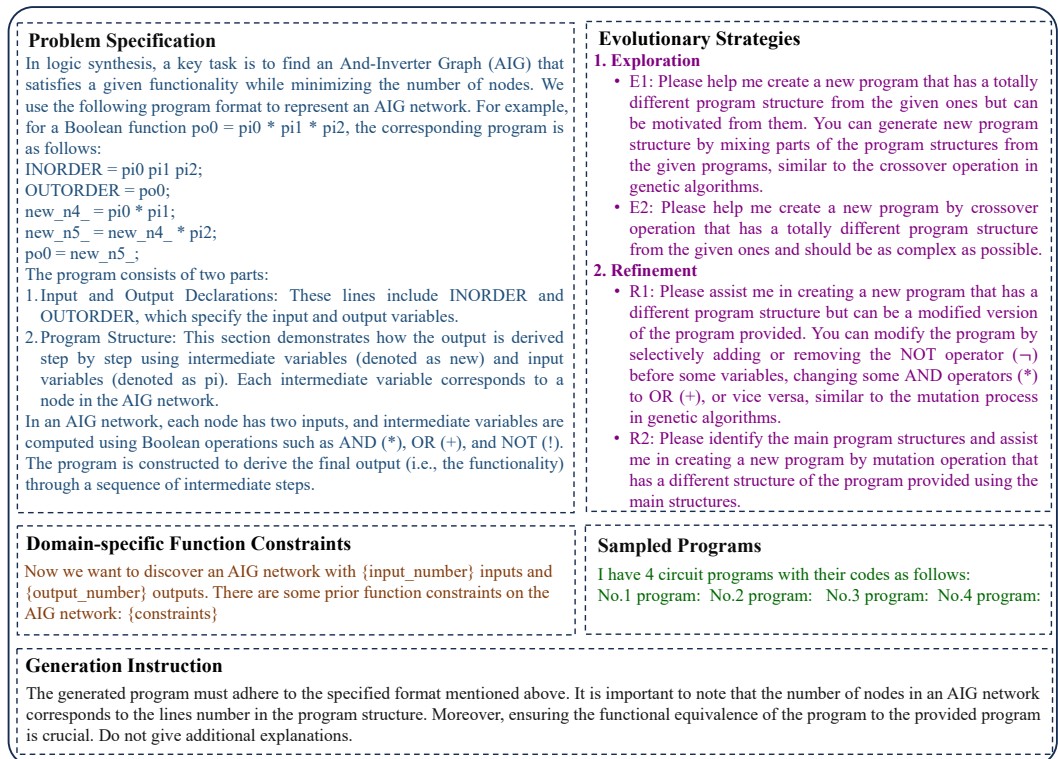

Figure 11: The prompts of CircutiEov. Specifically, the prompts consist of four main parts: Problem Specification, Circuit Program Examples, Evolutionary strategies (including exploration and refinement), and Domain-specific function constraints.

Table 6: We compare the circuit size and inference cost between our GPT-3.5-turbo and GPT-4o based approaches on the Random circuit benchmark. The results show that although GPT-3.5-turbo incurs a slight increase in size compared to GPT-4o, its cost is five times lower.

| Benchmark | SOTA Baseline | Ours (GPT-3.5-turbo) | Ours (GPT-3.5-turbo) | Ours (GPT-4o) | Ours (GPT-4o) |
|---|---|---|---|---|---|
| Circuit | Init Node ↓ | Init Node ↓ | Cost ($) ↓ | Init Node ↓ | Cost ($) ↓ |
| Random1 | 41 | 41 | **0.14** | **38** | 0.38 |
| Random2 | 37 | 36 | **0.11** | **33** | 0.41 |
| Random3 | 75 | **60** | **0.08** | **60** | 0.69 |
| Random4 | 152 | **138** | **0.18** | 140 | 0.52 |
| Random5 | 117 | **105** | **0.21** | **105** | 0.86 |
| Random6 | 1256 | 1110 | **0.11** | **1072** | 1.63 |
| Average | **279.67** | **248.33** | **0.15** | **241.33** | **0.75** |

Table 7: The results demonstrate that CircuitEvo can generate accurate circuits for all of the selected circuits from four widely used benchmarks.

| Benchmark | | | Boolformer | SPL | DSR | ICSR | DNAS | CircuitEvo (Ours) | | |
|---|---|---|---|---|---|---|---|---|---|---|
| Circuit | PI | PO | Acc(%) ↑ | Acc(%) ↑ | Acc(%) ↑ | Acc(%) ↑ | Acc(%) ↑ | Qwen2.5-7B | Deepseek-V3 | GPT-3.5-turbo |
| Arithmetic1 | 10 | 5 | 94.8 | 94.8 | 92.7 | 88.9 | 100.0 | 100.0 | 100.0 | 100.0 |
| Arithmetic2 | 16 | 6 | 75.3 | 83.5 | 77.4 | 55.7 | 100.0 | 100.0 | 100.0 | 100.0 |
| Arithmetic3 | 16 | 7 | 97.5 | 97.6 | 97.2 | 77.0 | 100.0 | 100.0 | 100.0 | 100.0 |
| LogicNets1 | 13 | 6 | 96.5 | 96.5 | 94.2 | 87.4 | 99.9 | 100.0 | 100.0 | 100.0 |
| LogicNets2 | 13 | 6 | 99.2 | 99.2 | 99.2 | 85.0 | 100.0 | 100.0 | 100.0 | 100.0 |
| LogicNets3 | 15 | 7 | 98.0 | 98.0 | 97.1 | 91.8 | 100.0 | 100.0 | 100.0 | 100.0 |
| Espresso1 | 11 | 3 | 94.0 | 94.9 | 94.5 | 94.0 | 100.0 | 100.0 | 100.0 | 100.0 |
| Espresso2 | 11 | 9 | 89.5 | 91.4 | 84.5 | 85.4 | 100.0 | 100.0 | 100.0 | 100.0 |
| Espresso3 | 12 | 17 | 88.4 | 88.5 | 88.5 | 87.9 | 99.9 | 100.0 | 100.0 | 100.0 |
| Espresso4 | 16 | 69 | 98.8 | 86.4 | 82.0 | 81.3 | 99.9 | 100.0 | 100.0 | 100.0 |
| Random1 | 7 | 2 | 87.5 | 87.5 | 87.5 | 82.8 | 100.0 | 100.0 | 100.0 | 100.0 |
| Random2 | 7 | 2 | 89.1 | 90.0 | 87.5 | 85.9 | 100.0 | 100.0 | 100.0 | 100.0 |
| Random3 | 10 | 3 | 87.5 | 90.0 | 86.5 | 86.7 | 100.0 | 100.0 | 100.0 | 100.0 |
| Random4 | 10 | 3 | 87.9 | 89.2 | 87.9 | 85.0 | 100.0 | 100.0 | 100.0 | 100.0 |
| Random5 | 12 | 3 | 86.2 | 88.4 | 86.8 | 86.3 | 100.0 | 100.0 | 100.0 | 100.0 |
| Random6 | 15 | 4 | 87.7 | 88.6 | 88.1 | 75.5 | 100.0 | 100.0 | 100.0 | 100.0 |
| **Average** | | | 91.1 | 91.5 | 89.5 | 83.5 | 100.0 | **100.0** | **100.0** | **100.0** |

Table 8: The optimization results (using *resyn2* operator for optimization) show that CircuitEvo achieves an average improvement of 11.09% in optimized circuit size over state-of-the-art baselines.

| Benchmark | | | Boolformer | SPL | DSR | ICSR | SOP | BDD | DNAS | CircuitEvo (Ours) | |
|---|---|---|---|---|---|---|---|---|---|---|---|
| Circuit | PI | PO | Opt Node ↓ | Opt Node ↓ | Opt Node ↓ | Opt Node ↓ | Opt Node ↓ | Opt Node ↓ | Opt Node ↓ | Opt Node ↓ | Impr.(%) ↑ |
| Arithmetic1 | 10 | 5 | 150 | 122 | 67 | 117 | **65** | **65** | 72 | **65** | 0.00 |
| Arithmetic2 | 16 | 6 | 244 | 157 | 70 | 230 | 51 | 55 | 105 | **44** | 13.73 |
| Arithmetic3 | 16 | 7 | 1360 | 1171 | 1108 | 1254 | 1178 | 1234 | 1161 | **973** | 12.18 |
| LogicNets1 | 13 | 6 | 141 | 115 | 109 | 214 | 127 | 117 | 160 | **105** | 3.67 |
| LogicNets2 | 13 | 6 | 75 | 75 | 75 | 78 | 83 | 95 | 92 | **48** | 36.00 |
| LogicNets3 | 15 | 7 | 196 | 207 | **154** | 190 | 184 | 199 | 227 | 186 | -20.78 |
| Espresso1 | 11 | 3 | 114 | 95 | 100 | 57 | **54** | **54** | 55 | **54** | 0.00 |
| Espresso2 | 11 | 9 | 203 | 220 | 179 | 187 | 140 | 131 | 121 | **85** | 29.75 |
| Espresso3 | 12 | 17 | 3199 | 2534 | 2865 | 2678 | 2572 | 2578 | 2565 | **2483** | 2.01 |
| Espresso4 | 16 | 69 | 623 | 1443 | 1456 | 1469 | 611 | 619 | 820 | **572** | 6.38 |
| Random1 | 7 | 2 | 42 | 38 | 39 | 41 | 38 | 39 | 37 | **35** | 5.41 |
| Random2 | 7 | 2 | 36 | 34 | 39 | **31** | 35 | 34 | 39 | 34 | -9.68 |
| Random3 | 10 | 3 | 81 | 68 | 99 | 88 | 100 | 91 | 64 | **30** | 53.13 |
| Random4 | 10 | 3 | 135 | 145 | 135 | 140 | 135 | 136 | 138 | **115** | 14.81 |
| Random5 | 12 | 3 | 218 | 283 | 171 | 168 | 156 | 106 | 105 | **76** | 27.62 |
| Random6 | 15 | 4 | 1695 | 1571 | 1405 | 1338 | 1290 | 1287 | 897 | **868** | 3.23 |
| **Average** | | | 532.00 | 517.38 | 504.44 | 517.50 | 426.19 | 427.50 | 416.13 | **360.81** | 11.09 |

Table 9: The optimization results (using *dc2* operator for optimization) show that CircuitEvo achieves an average improvement of 9.91% in optimized circuit size over state-of-the-art baselines.

| Benchmark | | | Boolformer | SPL | DSR | ICSR | SOP | BDD | DNAS | CircuitEvo (Ours) | |
|---|---|---|---|---|---|---|---|---|---|---|---|
| Circuit | PI | PO | Opt Node ↓ | Opt Node ↓ | Opt Node ↓ | Opt Node ↓ | Opt Node ↓ | Opt Node ↓ | Opt Node ↓ | Opt Node ↓ | Impr.(%) ↑ |
| Arithmetic1 | 10 | 5 | 137 | 113 | **62** | 112 | 63 | 64 | 73 | 63 | -1.61 |
| Arithmetic2 | 16 | 6 | 242 | 176 | 74 | 204 | 48 | 48 | 101 | **43** | 10.42 |
| Arithmetic3 | 16 | 7 | 1308 | 1145 | 1081 | 1217 | 1140 | 1194 | 1132 | **955** | 11.66 |
| LogicNets1 | 13 | 6 | 136 | 115 | 113 | 218 | 118 | 114 | 160 | **101** | 10.62 |
| LogicNets2 | 13 | 6 | 61 | 62 | 61 | 64 | 79 | 96 | 91 | **46** | 24.59 |
| LogicNets3 | 15 | 7 | 195 | 199 | **155** | 183 | 178 | 202 | 218 | 176 | -13.55 |
| Espresso1 | 11 | 3 | 108 | 86 | 89 | 54 | **53** | **53** | 54 | 55 | -3.77 |
| Espresso2 | 11 | 9 | 201 | 219 | 162 | 174 | 135 | 138 | 121 | **81** | 33.06 |
| Espresso3 | 12 | 17 | 3115 | 2478 | 2793 | 2611 | 2512 | 2515 | 2517 | **2418** | 2.42 |
| Espresso4 | 16 | 69 | 585 | 1376 | 1408 | 1388 | 589 | 577 | 820 | **554** | 3.99 |
| Random1 | 7 | 2 | 41 | 38 | 41 | 40 | **35** | 38 | 37 | **35** | 0.00 |
| Random2 | 7 | 2 | 34 | 33 | **29** | 33 | 35 | 31 | 30 | 32 | -10.34 |
| Random3 | 10 | 3 | 82 | 69 | 97 | 85 | 91 | 87 | 61 | **29** | 52.46 |
| Random4 | 10 | 3 | 128 | 143 | 126 | 139 | 129 | 134 | 142 | **116** | 7.94 |
| Random5 | 12 | 3 | 199 | 271 | 165 | 163 | 145 | 99 | 103 | **67** | 32.32 |
| Random6 | 15 | 4 | 1641 | 1536 | 1348 | 1289 | 1238 | 1256 | **854** | 868 | -1.64 |
| **Average** | | | 513.31 | 503.69 | 487.75 | 498.38 | 411.75 | 415.38 | 407.13 | **352.44** | 9.91 |

Table 10: The optimization results (using *compress2* operator for optimization) show that CircuitEvo achieves an average improvement of 10.79% in optimized circuit size over state-of-the-art baselines.

| Benchmark | | | Boolformer | SPL | DSR | ICSR | SOP | BDD | DNAS | CircuitEvo (Ours) | |
|---|---|---|---|---|---|---|---|---|---|---|---|
| Circuit | PI | PO | Opt Node ↓ | Opt Node ↓ | Opt Node ↓ | Opt Node ↓ | Opt Node ↓ | Opt Node ↓ | Opt Node ↓ | Opt Node ↓ | Impr.(%) ↑ |
| Arithmetic1 | 10 | 5 | 150 | 120 | 67 | 117 | **65** | **65** | 72 | 66 | -1.54 |
| Arithmetic2 | 16 | 6 | 233 | 151 | 50 | 222 | **43** | **43** | 105 | **43** | 0.00 |
| Arithmetic3 | 16 | 7 | 1360 | 1169 | 1105 | 1249 | 1177 | 1228 | 1162 | **975** | 11.76 |
| LogicNets1 | 13 | 6 | 146 | 113 | 110 | 211 | 120 | 117 | 160 | **103** | 6.36 |
| LogicNets2 | 13 | 6 | 75 | 75 | 75 | 78 | 83 | 94 | 93 | **48** | 36.00 |
| LogicNets3 | 15 | 7 | 199 | 206 | **154** | 190 | 183 | 197 | 227 | 185 | -20.13 |
| Espresso1 | 11 | 3 | 111 | 94 | 102 | 53 | **53** | **53** | 55 | **53** | 0.00 |
| Espresso2 | 11 | 9 | 198 | 220 | 174 | 187 | 137 | 130 | 121 | **84** | 30.58 |
| Espresso3 | 12 | 17 | 3195 | 2523 | 2860 | 2677 | 2570 | 2576 | 2564 | **2484** | 1.55 |
| Espresso4 | 16 | 69 | 631 | 1432 | 1444 | 1468 | 611 | 613 | 816 | **573** | 6.22 |
| Random1 | 7 | 2 | 42 | 38 | 39 | 41 | 36 | 39 | 37 | **35** | 2.78 |
| Random2 | 7 | 2 | 37 | 34 | 37 | **33** | 34 | **33** | 39 | **33** | 0.00 |
| Random3 | 10 | 3 | 81 | 68 | 98 | 90 | 102 | 91 | 64 | **30** | 53.13 |
| Random4 | 10 | 3 | 133 | 146 | 133 | 141 | 133 | 131 | 138 | **115** | 12.21 |
| Random5 | 12 | 3 | 215 | 284 | 171 | 165 | 152 | 105 | 104 | **72** | 30.77 |
| Random6 | 15 | 4 | 1700 | 1570 | 1403 | 1336 | 1287 | 1289 | **894** | 868 | 2.91 |
| **Average** | | | 531.63 | 515.19 | 501.38 | 516.13 | 424.13 | 425.25 | 415.69 | **360.44** | 10.79 |

Table 11: The logic sharing results demonstrate that our CircuitEvo significantly outperforms all Boolean function learning baselines in terms of the number of shared logic components.

| Benchmark | | | Boolformer | SPL | DSR | ICSR | CircuitEvo (Ours) |
|---|---|---|---|---|---|---|---|
| Circuit | PI | PO | Sharing | Sharing | Sharing | Sharing | Sharing |
| Arithmetic1 | 10 | 5 | 7 | 22 | 8 | 6 | 39 |
| Arithmetic2 | 16 | 6 | 13 | 13 | 13 | 14 | 42 |
| Arithmetic3 | 16 | 7 | 52 | 66 | 51 | 35 | 75 |
| LogicNets1 | 13 | 6 | 126 | 125 | 118 | 92 | 62 |
| LogicNets2 | 13 | 6 | 2 | 8 | 2 | 6 | 31 |
| LogicNets3 | 15 | 7 | 89 | 59 | 70 | 57 | 282 |
| Espresso1 | 11 | 3 | 58 | 79 | 67 | 69 | 109 |
| Espresso2 | 11 | 9 | 4 | 5 | 19 | 40 | 70 |
| Espresso3 | 12 | 17 | 2221 | 0 | 1556 | 1663 | 2384 |
| Espresso4 | 16 | 69 | 265 | 439 | 449 | 423 | 3998 |
| Random1 | 7 | 2 | 0 | 7 | 0 | 0 | 2 |
| Random2 | 7 | 2 | 0 | 1 | 1 | 0 | 2 |
| Random3 | 10 | 3 | 0 | 5 | 0 | 1 | 20 |
| Random4 | 10 | 3 | 9 | 14 | 5 | 7 | 9 |
| Random5 | 12 | 3 | 98 | 75 | 59 | 79 | 275 |
| Random6 | 15 | 4 | 28 | 33 | 23 | 29 | 15 |
| **Average** | | | 185.75 | 59.44 | 152.56 | 157.56 | **463.44** |

Table 12: The results show that the number of nodes in five circuits decreases as the circuit level increases, illustrating the triangular structure of the circuits generated by our method.

| Circuits | Arithmetic1 | | LogicNets1 | | LogicNets3 | | Espresso1 | | Random4 | |
|---|---|---|---|---|---|---|---|---|---|---|
| Lev | Node | Rate | Node | Rate | Node | Rate | Node | Rate | Node | Rate |
| 1 | 14 | 0.19 | 31 | 0.22 | 44 | 0.21 | 8 | 0.11 | 29 | 0.21 |
| 2 | 13 | 0.18 | 25 | 0.18 | 41 | 0.19 | 10 | 0.14 | 30 | 0.22 |
| 3 | 10 | 0.14 | 17 | 0.12 | 30 | 0.14 | 10 | 0.14 | 19 | 0.14 |
| 4 | 9 | 0.12 | 13 | 0.09 | 24 | 0.11 | 5 | 0.07 | 18 | 0.13 |
| 5 | 7 | 0.09 | 10 | 0.07 | 18 | 0.09 | 4 | 0.06 | 13 | 0.09 |
| 6 | 6 | 0.08 | 9 | 0.06 | 13 | 0.06 | 4 | 0.06 | 10 | 0.07 |
| 7 | 4 | 0.05 | 7 | 0.05 | 10 | 0.05 | 4 | 0.06 | 5 | 0.04 |
| 8 | 4 | 0.05 | 6 | 0.04 | 9 | 0.04 | 4 | 0.06 | 4 | 0.03 |
| 9 | 1 | 0.01 | 5 | 0.04 | 7 | 0.03 | 4 | 0.06 | 3 | 0.02 |
| 10 | 1 | 0.01 | 4 | 0.03 | 4 | 0.02 | 4 | 0.06 | 3 | 0.02 |
| 11 | 1 | 0.01 | 3 | 0.02 | 4 | 0.02 | 4 | 0.06 | 1 | 0.01 |
| 12 | 1 | 0.01 | 3 | 0.02 | 3 | 0.01 | 4 | 0.06 | 1 | 0.01 |
| 13 | 2 | 0.03 | 2 | 0.01 | 2 | 0.01 | 3 | 0.04 | 1 | 0.01 |
| 14 | 1 | 0.01 | 3 | 0.02 | 1 | 0.00 | 1 | 0.01 | 1 | 0.01 |
| 15 | 0 | 0.00 | 1 | 0.01 | 1 | 0.00 | 1 | 0.01 | 0 | 0.00 |
| 16 | 0 | 0.00 | 1 | 0.01 | 0 | 0.00 | 1 | 0.01 | 0 | 0.00 |

Table 13: This table provides further evidence of the triangular structure in our generated circuits. Specifically, the level rate, defined as half the ratio of the number of nodes at the bottom level to those at the top level, exceeds 1 in all cases, indicating a greater concentration of nodes at the bottom.

| Benchmark | | | CircuitEvo (Ours) | | Benchmark | | | CircuitEvo (Ours) | |
|---|---|---|---|---|---|---|---|---|---|
| Circuit | PI | PO | Lev | Lev rate | Circuit | PI | PO | Lev | Lev rate |
| Arithmetic1 | 10 | 5 | 14 | 5.73 | Espresso3 | 12 | 17 | 21 | 9.66 |
| Arithmetic2 | 16 | 6 | 12 | 4.37 | Espresso4 | 16 | 69 | 21 | 11.75 |
| Arithmetic3 | 16 | 7 | 24 | 18.45 | Random1 | 7 | 2 | 9 | 3.00 |
| LogicNets1 | 13 | 6 | 16 | 5.36 | Random2 | 7 | 2 | 9 | 3.00 |
| LogicNets2 | 13 | 6 | 17 | 2.57 | Random3 | 10 | 3 | 9 | 5.00 |
| LogicNets3 | 15 | 7 | 15 | 5.81 | Random4 | 10 | 3 | 14 | 8.86 |
| Espresso1 | 11 | 3 | 16 | 2.23 | Random5 | 12 | 3 | 11 | 5.18 |
| Espresso2 | 11 | 9 | 10 | 2.28 | Random6 | 15 | 4 | 23 | 15.32 |

Table 14: The Computation cost per circuit when using the GPT-3.5-turbo API as the backbone.

| Benchmark | | | CircuitEvo (Ours) | Benchmark | | | CircuitEvo (Ours) |
|---|---|---|---|---|---|---|---|
| Circuit | PI | PO | Cost ($) | Circuit | PI | PO | Cost ($) |
| Arithmetic1 | 10 | 5 | 0.09 | Espresso3 | 12 | 17 | 0.45 |
| Arithmetic2 | 16 | 6 | 0.16 | Espresso4 | 16 | 69 | 0.43 |
| Arithmetic3 | 16 | 7 | 0.27 | Random1 | 7 | 2 | 0.08 |
| LogicNets1 | 13 | 6 | 0.12 | Random2 | 7 | 2 | 0.11 |
| LogicNets2 | 13 | 6 | 0.20 | Random3 | 10 | 3 | 0.14 |
| LogicNets3 | 15 | 7 | 0.14 | Random4 | 10 | 3 | 0.11 |
| Espresso1 | 11 | 3 | 0.11 | Random5 | 12 | 3 | 0.18 |
| Espresso2 | 11 | 9 | 0.12 | Random6 | 15 | 4 | 0.21 |

