# OpenReview forum: "Evolving Graph Structured Programs for Circuit Generation with Large Language Models"
_ICLR.cc/2026/Conference — ICLR 2026 Poster_

### Official Review · Reviewer_m5UB · 2025-10-28

**Soundness:** 3
**Presentation:** 4
**Contribution:** 2
**Rating:** 6
**Confidence:** 4

**Summary:**

CircuitEvo is a novel framework that utilizes large language models (LLMs) to iteratively evolve circuit programs, enhancing compactness while ensuring functional accuracy. ​

The paper introduces CircuitEvo, the first LLM-based logic synthesis method that improves circuit compactness and functional accuracy. ​It employs a graph-structured circuit program representation to facilitate LLM comprehension and generation. ​It integrates an evolutionary program generator and a structure-aware function optimizer in an iterative framework. ​Empirical studies show  an average improvement of 6.74% in circuit size over state-of-the-art methods.


Task:  Given a truth table T , CircuitEvo aims to generate a circuit program P that exactly satisfies the given functionality while minimizing the number of structure lines (circuit nodes).

**Strengths:**

The proposed graph-structured circuit program effectively encodes circuit topology and functionality in a textual format. ​

The program consists of three parts: IO definition, structure description, and function definition.
This representation bridges the gap between circuit graphs and language-based reasoning, enhancing LLM capabilities. ​
The LS problem is formulated to generate a circuit program that satisfies functionality while minimizing the number of nodes. ​

Evolutionary Program Generation Process
CircuitEvo employs an evolutionary framework to generate diverse and promising circuit programs using LLMs. ​

Initializes a diverse population of accurate circuit programs through a decomposition strategy.
Utilizes domain-specific evolutionary prompting strategies to guide LLMs in generating high-quality programs. ​
Incorporates fitness-based population management to select the top-performing programs for subsequent iterations.

**Weaknesses:**

The LS summary uses very dated algorithms: existing LS approaches typically follow a generate-then optimize framework: they first generate a circuit from the truth table, then apply various operators for circuit optimization. Please update the references to show that topology-driven modern methods exist.

In terms of baselines I am curious to understand why you selected BDD (quite outdated---1993), and SOP (2017, also not SoA).
Further, since you use evolutionary methods, why not compare against one of the many evolutionary SoA methods; some very recent references are below. The failure to compare against existing evolutionary SoA methods means that we can't evaluate ablation techniques well, i.e., is it the evolutionary part that makes the difference, the graph part, or what?


Chen, Yu, and Shao-Yun Fang. "Optimizing Analog Circuit Design Through a Machine Learning‐Assisted Evolutionary Algorithm." Electronics Letters 61, no. 1 (2025): e70331.

Campilho-Gomes, Miguel, Rui Tavares, and João Goes. "Analog flat-level circuit synthesis with genetic algorithms." IEEE Access (2024).

Srikanth, V., P. Aswini, Rakesh Chandrashekar, N. Sirisha, Manish Kumar, and K. Adnan. "Machine Learning-Based Analogue Circuit Design for Stage Categorization and Evolutionary Optimization." In 2024 Second International Conference Computational and Characterization Techniques in Engineering & Sciences (IC3TES), pp. 1-6. IEEE, 2024.

**Questions:**

1. please address the choice of baselines
2. please carefully focus on ablation regarding which aspect of your approach provides the real benefit (see comments about choice of baseline)

---

> ### Author Response · Authors · 2025-11-22
> **Response to Reviewer m5UB---Part 1/2**
>
> Dear Reviewer m5UB,
>
> We greatly appreciate your insightful comments and thoughtful recognition of our work! We sincerely hope that our responses could properly address all your concerns and help strengthen your confidence in our submission. If so, we would deeply appreciate it if you could raise your score. If not, please let us know your further concerns, and we will continue actively responding to your comments and improving our submission.
>
> ### Weakness 1
> > **The LS summary uses very dated algorithms: existing LS approaches typically follow a generate-then optimize framework: they first generate a circuit from the truth table, then apply various operators for circuit optimization. Please update the references to show that topology-driven modern methods exist.**
>
> We sincerely thank the reviewer for the valuable comments. Following your suggestions, **we have updated the references in the revised manuscript (line 43)** to include recent topology-driven LS methods and modern references, including **[1-3]** to ensure a more complete and up-to-date literature coverage.
>
> Once again, we would like to appreciate the reviewer for their careful and insightful suggestions.
>
> [1]. Wang, Zhihai, et al. Towards next-generation logic synthesis: A scalable neural circuit generation framework. NeurIPS 2024.
>
> [2]. Cheng, Shuyao, et al. "Automated CPU design by learning from input-output examples." IJCAI 2024.
>
> [3]. Li X, Li X, Chen L, et al. Circuit Transformer: A Transformer That Preserves Logical Equivalence. ICLR 2025.
> ### Weakness 2 and Question 1
> > **In terms of baselines I am curious to understand why you selected BDD (quite outdated---1993), and SOP (2017, also not SoA). Further, since you use evolutionary methods, why not compare against one of the many evolutionary SoA methods?**
>
> We sincerely thank the reviewer for the insightful questions. First, we would like to clarify that **SOP and BDD are two widely used industrial operators** implemented in the open-source logic synthesis tool **ABC**. Even in the most recent circuit generation competitions such as IWLS [4], these operators are adopted as official baselines owing to their **efficiency and effectiveness** in practical synthesis flows.
>
> Regarding evolutionary approaches, to the best of our knowledge, **no existing evolutionary method directly targets the (combinational) circuit generation task**. To provide a fair comparison, we therefore develop an evolutionary baseline based on **linear genetic programming (LGP) [5]**, which evaluates and evolves circuit programs using genetic programming methods. As shown in Table a, our method significantly outperforms this evolutionary baseline, highlighting the strong generation capability of the LLM and the effectiveness of integrating evolutionary search within our framework.
>
> **Table a:** Comparison results between our method and a linear genetic programming (LGP) based baseline.
> |             | LGP        | CircuitEvo | CircuitEvo-LGP |
> | ----------- | ---------- | ---------- | -------------- |
> | Circuit     | Init Node  | Init Node  | Impr(%)        |
> | Arithmetic1 | 78         | 74         | **5.13**       |
> | Arithmetic2 | 98         | 102        | -**4.08**      |
> | Arithmetic3 | 1229       | 1206       | **1.87**       |
> | Espresso1   | 71         | 71         | **0.00**       |
> | Espresso2   | 145        | 128        | **11.72**      |
> | Espresso3   | 3220       | 3184       | **1.12**       |
> | Espresso4   | 940        | 848        | **9.79**       |
> | LogicNets1  | 139        | 139        | **0.00**       |
> | LogicNets2  | 75         | 74         | **1.33**       |
> | LogicNets3  | 243        | 211        | **13.17**      |
> | Random1     | 44         | 41         | **6.82**           |
> | Random2     | 37         | 36         | **2.70**       |
> | Random3     | 68         | 60         | **11.76**      |
> | Random4     | 147        | 138        | **6.12**          |
> | Random5     | 110        | 105        | **4.55**       |
> | Random6     | 1110       | 1110       | **0.00**           |
> | **Average** | **484.63** | **470.44**     | **4.50**       |
>
> [4]. Rai S, Neto W L, Miyasaka Y, et al. Logic synthesis meets machine learning: Trading exactness for generalization. 2021 Design, Automation & Test in Europe Conference & Exhibition (DATE).
>
> [5]. Oltean M, Grosan C. A comparison of several linear genetic programming techniques. Complex Systems, 2007.

---

> ### Author Response · Authors · 2025-11-22
> **Response to Reviewer m5UB---Part 2/2**
>
> ### Weakness 3 and Question 2
> > **please carefully focus on ablation regarding which aspect of your approach provides the real benefit (see comments about choice of baseline)**
>
> We sincerely thank the reviewer for the thoughtful suggestion. In response, we conducted detailed ablation studies to isolate the contribution of each component in our framework. The results in Table 4 in the manuscript show that **every major module---LLM-based generation, evolutionary search strategies, local search, and the function optimizer---provides a clear and measurable benefit**. For your convenience, we present the Table 4 as follows. Specifically, removing any individual component leads to a degradation in circuit compactness or functional accuracy, demonstrating that the performance improvements arise from the complementary strengths of all parts rather than a single dominant factor.
>
> **Table 4 (from the manuscript)**: The ablation results show that each component plays an important role in our CircuitEvo.
>
> |  Benchmark  |  CircuitEvo (Ours) |                        |       w/o LLM      |                        |     w/o Program    |                        |    w/o Evolution   |                        |      w/o Local     |                        |   w/o Completion   |                        |
> |:-----------:|:------------------:|:----------------------:|:------------------:|:----------------------:|:------------------:|:----------------------:|:------------------:|:----------------------:|:------------------:|:----------------------:|:------------------:|:----------------------:|
> |   Circuit   | Acc(\%) | Node | Acc(\%) | Node | Acc(\%) | Node | Acc(\%) | Node | Acc(\%) | Node | Acc(\%) | Node |
> | Arithmetic1 |        100.0       |           74           |        100.0       |           78           |        100.0       |           81           |        100.0       |           78           |        100.0       |           79           |        100.0       |           74           |
> | Arithmetic3 |        100.0       |          1206          |        100.0       |          1229          |        100.0       |          1229          |        100.0       |          1233          |        100.0       |          1178          |        67.8        |           27           |
> |  Espresso3  |        100.0       |          3184          |        100.0       |          3220          |        100.0       |          3377          |        100.0       |          3184          |        100.0       |          3337          |        82.6        |           200          |
> |  Espresso4  |        100.0       |           848          |        100.0       |           940          |        100.0       |           940          |        100.0       |           911          |        100.0       |           911          |        86.4        |            9           |
> |  LogicNets1 |        100.0       |           139          |        100.0       |           140          |        100.0       |           148          |        100.0       |           150          |        100.0       |           140          |        70.1        |            2           |
> |  LogicNets2 |        100.0       |           74           |        100.0       |           90           |        100.0       |           74           |        100.0       |           75           |        100.0       |           75           |        60.6        |            3           |
> |   Random4   |        100.0       |           138          |        100.0        |           147          |        100.0       |           146          |        100.0       |           151          |        100.0       |           144          |        84.5        |           20           |
> |   Random5   |        100.0       |           105          |        100.0       |           110          |        100.0       |           106          |        100.0       |           106          |        100.0       |           105          |        83.3        |           49           |

---

### Official Review · Reviewer_8gZq · 2025-10-29

**Soundness:** 2
**Presentation:** 3
**Contribution:** 2
**Rating:** 4
**Confidence:** 4

**Summary:**

This paper introduces CircuitEvo, an LLM-based evolutionary framework for circuit programs, to tackle the difficulty in balancing compactness and accuracy in LS. By leveraging a graph-based program representation alongside LLM-driven evolution and structure-aware optimization, it iteratively optimizes sequential circuits. Results show 100% generation accuracy, an average 6.74% reduction in circuit size, and significantly better search efficiency versus baseline methods.

**Strengths:**

1. This paper addresses the important problem of automated sequential circuit design, offering well-defined practical value.
2. The experimental results significantly outperform existing baseline methods, demonstrating clear advantages in terms of accuracy, scale, and efficiency.
3. The theoretical derivations are logically rigorous, with proofs provided for the relevant remarks.

**Weaknesses:**

1. The paper has a deficiency in terms of innovation. The core techniques—LLM generation, evolutionary algorithms, and circuit function completion—constitute an integration of existing methods, and no novel model architecture or theoretical framework is proposed. Furthermore, the local program search is essentially an incremental improvement of a greedy strategy, lacking a breakthrough in algorithmic originality.
2. The experimental evaluation in the paper is limited to synchronous sequential circuits and combinational circuits. It does not demonstrate the framework's applicability to asynchronous sequential circuits or circuits with complex feedback loops.
3. The paper lacks comprehensive ablation studies. It only examines the impact of variants like removing the LLM or the evolutionary algorithm, but fails to evaluate the individual contribution of different evolutionary strategies (E1/E2/R1/R2). For instance, how much would circuit diversity decrease if the E1 exploration strategy were removed?
4. Validation on extreme scales is missing. The framework was not tested on very large-scale circuits (e.g., with inputs >16 and outputs >70), leaving its scalability unverified.
5. The comparative analysis of LLMs is brief. It merely lists results from models like GPT-3.5-turbo and Deepseek-V3 without analyzing the impact of LLM parameter scale (e.g., 7B vs. 13B) on generation quality.

**Questions:**

1. Please respond to the concerns I have raised in the 'weaknesses' section. If the revision can adequately address most of the critical issues, I would consider raising the score.
2. In my personal opinion, the contribution of this paper lies more in its solid experimentation and significant performance improvements, rather than in its theoretical advancement in AI. Considering this, EDA-focused conferences (such as DAC or ISCA, with an upcoming November deadline) would be a more suitable venue for it.

---

> ### Author Response · Authors · 2025-11-22
> **Response to Reviewer 8gZq---Part 1/4**
>
> Dear Reviewer 8gZq,
>
> We greatly appreciate your careful reading and constructive feedback! We have provided further responses as follows. We sincerely hope that our responses could properly address all your concerns. If so, we would deeply appreciate it if you could raise your score. If not, please let us know your further concerns, and we will continue actively responding to your comments and improving our submission.
>
> ### Weakness 1
> >  **The paper has a deficiency in terms of innovation. The core techniques—LLM generation, evolutionary algorithms, and circuit function completion—constitute an integration of existing methods, and no novel model architecture or theoretical framework is proposed. Furthermore, the local program search is essentially an incremental improvement of a greedy strategy, lacking a breakthrough in algorithmic originality.**
>
> We sincerely thank the reviewer for the opportunity to clarify the innovation of our work. The major novelty of method lies in:
> - **A novel graph-structured circuit program representation that first bridges the gap between circuit graph structures and language-based reasoning**, enabling LLMs to effectively understand and manipulate circuit topologies;
> - **A dedicated framework that achieves the first successful application of large language models to circuit generation**, integrating global LLM generation with structure-aware optimization to reliably synthesize compact and functionally correct circuits.
>
> Specifically, while using LLMs as candidate generator inside an evolutionary search framework has already been well studied within the community on different problems [1][2][3], these framework cannot generalize to Logic Synthesis (LS) tasks due to two key challenges: 1) **Lack of LLM-aware circuit representations;** 2) **Difficulty maintaining both structure compactness and functional correctness.** To address the first challenge, our graph-structured circuit program providing an LLM-friendly formulation that generalizes across multiple circuit types. To address the second challenge, we introduce a LLM-based evolutionary framework which effectively enables LLM's to evolve circuits toward greater compactness while keeping functional correctness. Finally, **all components of our framework are carefully designed with domain-specific insights from logic synthesis**, which is crucial for enabling LLMs to successfully tackle the circuit generation problem.
>
> [1]. Liu F, Xialiang T, Yuan M, et al. Evolution of Heuristics: Towards Efficient Automatic Algorithm Design Using Large Language Model. ICML, 2024.
>
> [2]. Guo P, Liu F, Lin X, et al. L-AutoDA: Leveraging Large Language Models for Automated Decision-based Adversarial Attacks. CoRR, 2024.
>
> [3]. Pluhacek M, Kazikova A, Kadavy T, et al. Leveraging large language models for the generation of novel metaheuristic optimization algorithms. Proceedings of the Companion Conference on Genetic and Evolutionary Computation, 2024.
>
> ### Weakness 2
> > **The experimental evaluation in the paper is limited to synchronous sequential circuits and combinational circuits. It does not demonstrate the framework's applicability to asynchronous sequential circuits or circuits with complex feedback loops.**
>
> We sincerely appreciate the reviewer for the valuable comments. We would like to clarify that **the circuit generation task in this work focuses exclusively on combinational logic circuit generation**. In recent years, numerous competitions---such as IWLS [4]---as well as prior research efforts have centered on this task, underscoring its relevance and difficulty. All benchmarks used in our experiments are standard combinational datasets from the IWLS contest. Moreover, we have evaluated our method on **one of the most challenging circuits from the IWLS contest, Espresso4, which contains 16 inputs and 69 outputs**, further demonstrating the capability and robustness of our approach.
>
> [4]. Rai S, Neto W L, Miyasaka Y, et al. Logic synthesis meets machine learning: Trading exactness for generalization. 2021 Design, Automation & Test in Europe Conference & Exhibition (DATE).

---

> ### Author Response · Authors · 2025-11-22
> **Response to Reviewer 8gZq---Part 2/4**
>
> ### Weakness 3
> > **The paper lacks comprehensive ablation studies. It only examines the impact of variants like removing the LLM or the evolutionary algorithm, but fails to evaluate the individual contribution of different evolutionary strategies (E1/E2/R1/R2). For instance, how much would circuit diversity decrease if the E1 exploration strategy were removed?**
>
> We sincerely thank the reviewer for the valuable comments. Following your suggestion, we conducted an ablation study on two types of evolutionary strategies, E1 and M1. As shown in Table a, **each of these strategies plays a critical role in the generation stage**, contributing significantly to the quality of the resulting circuits.
>
> **Table a:** We conducted an ablation study on four different challenging circuits to evaluate two evolutionary strategies E1 and M1.
> | Method       | CircuitEvo | CircuitEvo w/o E1 | CircuitEvo w/o M1 |
> |--------------|------------|-------------------|-------------------|
> | Circuit      | Init Nd    | Init Nd           | Init Nd           |
> | Arithmetic3  | **1206**       | 1212              | 1228              |
> | Espresso1    | **848**        | 911               | 911               |
> | LogicNets2   | **74**         | 79                | 83                |
> | Random5      | **138**        | 147               | 145               |

---

> ### Author Response · Authors · 2025-11-22
> **Response to Reviewer 8gZq---Part 3/4**
>
> ### Weakness 4
> > **Validation on extreme scales is missing. The framework was not tested on very large-scale circuits (e.g., with inputs >16 and outputs >70), leaving its scalability unverified.**
>
> We sincerely thank the reviewer for the insightful comments. Following your suggestions, we conducted a through experiments to analyze the scaling behavior of our methods.
>
> Specifically, we randomly generate lots of circuits by varying the number of inputs from 16 to 20 and the number of outputs from 1 to 100. The results in Table b demonstrate that **our method with GPT-3.5-turbo exhibits a substantial scalability advantage over the state-of-the-art machine learning approach DNAS**. Specifically, our method can **successfully generalize to 18 inputs and 100 outputs with about 10000 nodes**. Motivated by the scalability results, we observe that **the primary limitation of our method arises from the increasing circuit size**, which leads to longer and more complex generated programs. To address this problem, we propose two potential solutions.
>
> Firstly, **leveraging large language models with substantially longer context windows** can significantly enhance the scalability of our approach. In fact, the state-of-the-art models support context lengths that are roughly 250 times larger than that of GPT-3.5-turbo. Once paired with these LLMs, our method has the potential to directly generate circuits at the hundred-thousand or even million scale.
>
> Secondly, for very large-scale circuits---where the corresponding programs are significantly longer than the context window length---**circuit decomposition** techniques can be employed to break down the circuit into smaller, more manageable sub-circuits that can be learned or optimized individually. For instance, [5] adopts **Shannon decomposition**, which partitions the original truth table into several sub-truth tables based on input variable conditioning. Each sub-truth table is independently synthesized and then combined to reconstruct the full circuit. In a different approach, [6] employs a Monte Carlo Tree Search (MCTS)-based sampling strategy, which **performs multiple rounds of input-output sampling** to construct sub-truth tables. These sub-components are subsequently synthesized and merged, enabling scalable synthesis for large and complex circuits.
>
>
> **Table b:** Comparative evaluation of our method and the state-of-the-art ML method DNAS in terms of scalability performance. Here, *NA* indicates that the method failed to generate a valid circuit within the rebuttal time limit.
> | PO=1       | PI=16     | PI=17     | PI=18     | PI=19     | PI=20     |
> | ---------- | --------- | --------- | --------- | --------- | --------- |
> | Method     | Init Nd   | Init Nd   | Init Nd   | Init Nd   | Init Nd   |
> | CircuitEvo | 436       | 626       | 901       | 1287      | 1850      |
> | DNAS       | 558       | NA        | NA        | NA        | NA        |
> | **PO=5**   | **PI=16** | **PI=17** | **PI=18** | **PI=19** | **PI=20** |
> | Method     | Init Nd   | Init Nd   | Init Nd   | Init Nd   | Init Nd   |
> | CircuitEvo | 1028      | 1444      | 2113      | 2944      | 4212      |
> | DNAS       | 1256      | NA        | NA        | NA        | NA        |
> | **PO=10**  | **PI=16** | **PI=17** | **PI=18** | **PI=19** | **PI=20** |
> | Method     | Init Nd   | Init Nd   | Init Nd   | Init Nd   | Init Nd   |
> | CircuitEvo | 1457      | 2055      | 2971      | 4188      | 5953      |
> | DNAS       | 1641      | NA        | NA        | NA        | NA        |
> | **PO=20**  | **PI=16** | **PI=17** | **PI=18** | **PI=19** | **PI=20** |
> | Method     | Init Nd   | Init Nd   | Init Nd   | Init Nd   | Init Nd   |
> | CircuitEvo | 2079      | 2939      | 4181      | 5955      | 8410      |
> | DNAS       | 2222      | NA        | NA        | NA        | NA        |
> | **PO=50**  | **PI=16** | **PI=17** | **PI=18** | **PI=19** | **PI=20** |
> | Method     | Init Nd   | Init Nd   | Init Nd   | Init Nd   | Init Nd   |
> | CircuitEvo | 3209      | 4629      | 6529      | 9292      | NA        |
> | DNAS       | 3689        | NA        | NA        | NA        | NA        |
> | **PO=100** | **PI=16** | **PI=17** | **PI=18** | **PI=19** | **PI=20** |
> | Method     | Init Nd   | Init Nd   | Init Nd   | Init Nd   | Init Nd   |
> | CircuitEvo | 4520      | 6393      | 9236      | NA        | NA        |
> | DNAS       | NA        | NA        | NA        | NA        | NA        |
>
> [5]. Wang, Zhihai, et al. Towards next-generation logic synthesis: A scalable neural circuit generation framework. NeurIPS 2024.
>
> [6]. Cheng, Shuyao, et al. "Automated CPU design by learning from input-output examples." IJCAI 2024.

---

> > ### Author Response · Authors · 2025-11-22
> > **Response to Reviewer 8gZq---Part 4/4**
> >
> > ### Weakness 5
> > > **The comparative analysis of LLMs is brief. It merely lists results from models like GPT-3.5-turbo and Deepseek-V3 without analyzing the impact of LLM parameter scale (e.g., 7B vs. 13B) on generation quality.**
> >
> > We sincerely thank the reviewer for the valuable comments.
> > Thanks for your insightful suggestions! To thoroughly investigate the impact of LLM parameter scale on circuit generation quality, we conducted additional comprehensive experiments with **Qwen2.5-7B and the larger Qwen2.5-14B** under the same experimental settings. Results in Table c demonstrate that **there exists a consistent performance improvement as the model scale increases from 7B to 14B**. Specifically, Qwen2.5-14B delivers better or comparable results on 10 out of 12 circuits, while reducing the average circuit size from 101.00 to 99.75.
> >
> > **Table c:** Comparison between Qwen2.5-7B and Qwen2.5-14B on the generation quality.
> > | Circuit     |  Qwen2.5-7B | Qwen2.5-14B |
> > |:-----------:|:----------:|:-----------:|
> > | Method | Init Nd     | Init Nd      |
> > | Arithmetic1 | 77     | **77**      |
> > | Arithmetic2 | **105**    | 114         |
> > | LogicNets1  | 149        | **140**         |
> > | LogicNets2  | 74     | **74**      |
> > | LogicNets3  | 212        | **211**         |
> > | Espresso1   | 71     | **71**      |
> > | Espresso2   | 133        | **128**     |
> > | Random1     | 42         | **38**      |
> > | Random2     | 36     | **36**      |
> > | Random3     | 68         | **60**      |
> > | Random4     | **140**    | 143         |
> > | Random5     | 105    | **105**     |
> > | Average     | 101.00     | **99.75**   |
> >
> > ### Question 1
> > > **Please respond to the concerns I have raised in the 'weaknesses' section. If the revision can adequately address most of the critical issues, I would consider raising the score.**
> >
> > We sincerely thank the reviewer for the thoughtful and constructive comments. The comments in the weaknesses section are **highly valuable for strengthening our work**, and we are carefully revising the manuscript to address each point in depth. **We truly appreciate your willingness to reconsider the score after the revision**. If there are any further questions or concerns, please don't hesitate to let us know, and we will continue actively responding to your comments and improving our submission.
> >
> > ### Question 2
> > > **In my personal opinion, the contribution of this paper lies more in its solid experimentation and significant performance improvements, rather than in its theoretical advancement in AI. Considering this, EDA-focused conferences (such as DAC or ISCA, with an upcoming November deadline) would be a more suitable venue for it.**
> >
> > We sincerely thank the reviewer for the recognition of our empirical contributions and for the thoughtful suggestion regarding suitable venues. We fully appreciate the perspective that our work delivers strong experimental results and practical performance gains.
> >
> > At the same time, we would like to humbly clarify the methodological contributions our work brings to the AI4EDA community. Specifically, our paper introduces:
> > - **A novel graph-structured circuit program representation** that enables LLMs to reason over circuit topologies in a way that was previously infeasible;
> > - **A dedicated framework that achieves the first successful application of large language models to circuit generation**, integrating global LLM generation with structure-aware optimization to reliably synthesize compact and functionally correct circuits.
> >
> > Overall, these contributions **provide a principled and generalizable methodology for applying large language models to circuit generation**. We also deeply appreciate the reviewer’s encouragement and humbly believe that our work **has the potential to make a valuable contribution to the broader AI4EDA research community**.

---

### Official Review · Reviewer_LswG · 2025-10-31

**Soundness:** 3
**Presentation:** 4
**Contribution:** 3
**Rating:** 8
**Confidence:** 2

**Summary:**

The paper proposes a framework for logic synthesis, CircuitEvo, which employs LLMs within an evolutionary loop to improve final circuit compactness while preserving functional accuracy. The framework is carefully designed to account for the specific characteristics of circuit synthesis and introduces several domain-tailored operations, such as population initialization via the Shannon decomposition theorem and domain-specific function constraints. These components appear both reasonable and effective in ensuring a diverse set of accurate initial solutions, or enhancing optimization efficiency under an exponentially large search space. Experiments conducted on multi-benchmarks, in comparison with multiple baselines, demonstrate the effectiveness and efficiency of CircuitEvo. Moreover, the authors report an interesting additional finding: triangular structures play a significant role in circuit compactness.

**Strengths:**

1. Although integrating LLMs into a circuit-design loop is not entirely novel, the authors propose several domain-specific operations to enhance how inherently uncertain LLMs can correctly and efficiently generate the desired circuit structures in this domain — which is commendable.
2. The paper is clearly written, with well-designed figures that aid understanding.
3, I think the authors have conducted comprehensive experiments (although I am not very familiar with experimental design in this field). The benchmarks and baselines appear sufficiently chosen, and the experimental results look impressive.

**Weaknesses:**

1. The experimental results are based on single-run evaluation. This limits the statistical significance of the findings, especially considering that LLMs can be highly sensitive to random factors such as initialization and prompt variations, whose influences on the experiments are not  discussed in the paper.

2.Several highly relevant off-the-shelf baselines were not included in the experimental comparison, such as DARTS- [1] and T-Net [2].

[1] Xiangxiang Chu, Xiaoxing Wang, Bo Zhang, Shun Lu, Xiaolin Wei, and Junchi Yan. Darts-: robustly stepping out of performance collapse without indicators. arXiv preprint arXiv:2009.01027, 2020.

[2]  Zhihai Wang, Jie Wang, Qingyue Yang, Yinqi Bai, Xing Li, Lei Chen, Jianye HAO, Mingxuan Yuan, Bin Li, Yongdong Zhang, and Feng Wu. Towards next-generation logic synthesis: A scalable neural circuit generation framework. In The Thirty-eighth Annual Conference on Neural Information Processing Systems, 2024.

**Questions:**

1. It seems hard to believe that the involvement of LLMs results in such a low convergence time of only 1.48 hours. I am curious whether the reported runtime excludes the LLM inference time between inputting prompt and generating output.
2. Could the authors provide a breakdown of the time consumption for each operation within the evolutionary program generation process in each iteration — for example, the proportions spent on crossover, mutation, and functionality completion?
3. There are some cases where CircuitEvo underperforms the baselines in terms of circuit compactness. Could the authors provide a clear explanation for these results?
4. I look forward to the clarifications that address the above weaknesses.

---

> ### Author Response · Authors · 2025-11-22
> **Response to Reviewer LswG---Part 1/2**
>
> Dear Reviewer LswG,
>
> We greatly appreciate your insightful comments and thoughtful recognition of our work! We sincerely hope that our responses could properly address all your concerns and help strengthen your confidence in our submission. If so, we would deeply appreciate it if you could raise your score (or confidence). If not, please let us know your further concerns, and we will continue actively responding to your comments and improving our submission.
>
> ### Weakness 1
> > **The experimental results are based on single-run evaluation. This limits the statistical significance of the findings, especially considering that LLMs can be highly sensitive to random factors such as initialization and prompt variations, whose influences on the experiments are not discussed in the paper.**
>
> We sincerely thank the reviewer for the constructive comments. Following your suggestions, we conducted **three fully independent runs with different random seeds** on the complete benchmark suite using GPT-3.5-turbo. As shown in Table a, **our method demonstrates robust performance**, with **an average standard deviation of 0.29**, corresponding to only **0.06% (0.29/470.86) of the average circuit size**.
>
>
> **Table a:** Results of three independent full runs with GPT-3.5-turbo.
>
> | Circuit     |Ours Avg ± Std |
> |:-----------:|:-------------:|
> | Arithmetic1 | 76.67 ± 1.89   |
> | Arithmetic2 | 106.67 ± 3.30  |
> | Arithmetic3 | 1202.67 ± 2.36 |
> | Espresso1   | 71.00 ± 0.00   |
> | Espresso2   | 128.00 ± 0.00  |
> | Espresso3   | 3184.00 ± 0.00 |
> | Espresso4   | 848.00 ± 0.00  |
> | LogicNets1  | 139.67 ± 0.47  |
> | LogicNets2  | 74.67 ± 0.47   |
> | LogicNets3  | 211.67 ± 0.47  |
> | Random1     | 43.67 ± 1.89   |
> | Random2     | 36.00 ± 0.00   |
> | Random3     | 65.33 ± 3.77   |
> | Random4     | 138.00 ± 0.00  |
> | Random5     | 105.67 ± 0.47  |
> | Random6     | 1102.00 ± 5.66 |
> | **Average**     | **470.86 ± 0.29**  |
>
> ### Weakness 2
> > **Several highly relevant off-the-shelf baselines were not included in the experimental comparison, such as DARTS- [1] and T-Net [2].**
>
> We sincerely thank the reviewer for the valuable suggestions. First, we would like to clarify that **T-Net [2] is identical to the state-of-the-art machine learning baseline “DNAS” reported throughout our manuscript**. Additionally, DARTS [1] represents another network architecture search method, which **we have now cited as a representative ML baseline in the revised version (line 364)**. We specifically compare our method against both of these baselines. As shown in Table b, our method **achieves substantially higher generation accuracy** than both approaches on four widely-used benchmarks. Furthermore, when compared with **the stronger baseline T-Net** in terms of generation quality, the results in Table 2 in the manuscript show that **CircuitEvo consistently produces smaller final circuits**, demonstrating the effectiveness of our method in generating compact circuits.
>
> **Table b:** Comparison results between our method and two network architecture search methods in terms of the generation accuracy.
>
> |   | DARTS | T-Net  | CircuitEvo (Ours)|
> |:-----------:|:----------:|:---------:|:---------:|
> | **Benchmark** | Acc(%) | Acc(%)  | Acc(%)  |
> | Arithmetic  | 81.50      | 100.00    | 100.00    |
> | Espresso    | 91.00      | 99.95     | 100.00    |
> | LogicNets   | 92.06      | 99.97     | 100.00    |
> | Random      | 98.73      | 100.00    | 100.00    |
> | **Average**  | **90.8** | **99.9** | **100.00** |

---

> ### Author Response · Authors · 2025-11-22
> **Response to Reviewer LswG---Part 2/2**
>
> ### Question 1
> > **It seems hard to believe that the involvement of LLMs results in such a low convergence time of only 1.48 hours. I am curious whether the reported runtime excludes the LLM inference time between inputting prompt and generating output.**
>
> We sincerely thank the reviewer for the careful and valuable comment. **The reported convergence time of 1.48 hours fully accounts for the complete LLM inference time**, including the duration from prompt submission to final token generation for all calls throughout the evolutionary search. For a detailed analysis of the runtime breakdown and the contribution of LLM inference to the overall time cost, please refer to *Question 3* in our rebuttal.
>
> ### Question 2
> > **Could the authors provide a breakdown of the time consumption for each operation within the evolutionary program generation process in each iteration — for example, the proportions spent on crossover, mutation, and functionality completion?**
>
> We sincerely thank the reviewer for the insightful suggestions. We provide a detailed breakdown of the time consumption across different operations in our evolutionary process. As shown in Table c, **LLM-based variation operations (E1 + E2 + M1 + M2) account for approximately 75–80% of the total runtime**. Among these, crossover operations (E1 + E2) contribute roughly **40–60%**, while mutation operations (M1 + M2) account for about **30–40%**. **Functionality optimization of LLM-generated candidates consumes approximately 15%** of the total time. The remaining runtime is spent on selection, evaluation, and other lightweight utilities.
>
> **Table c:** Detailed Time breakdown results of CircuitEvo on four representative circuits reported in the manuscript.
> | | E1 | E2 | M1 | M2 | Optimization |Others |
> |:-----------:|:------:|:------:|:------:|:------:|:--------------:|:-----------------:|
> | Circuit   | Ratio (%) | Ratio (%) | Ratio (%) | Ratio (%) | Ratio (%)|Ratio (%)  |
> | Arithmetic1 | 15.7   | 23.13  | 17.94  | 20.86  | 14.14  | 8.23       |
> | LogicNets1  | 26.77  | 15.14  | 18.09  | 19.88  | 14.25  | 5.85       |
> | Espresso1   | 20.6   | 27.49  | 15.97  | 15.35  | 13.02  | 7.57       |
> | Random5     | 19.25  | 24.17  | 19.49  | 14.82  | 12.99  | 9.28       |
>
> ### Question 3
> > **There are some cases where CircuitEvo underperforms the baselines in terms of circuit compactness. Could the authors provide a clear explanation for these results?**
>
> We sincerely thank the reviewer for the thoughtful question. We acknowledge that on LogicNets3, CircuitEvo underperforms the DSR baseline in terms of circuit compactness. This gap primarily arises because **DSR happens to discover a particularly compact Boolean function representation for this specific circuit**, which——after functional optimization——leads to a smaller final implementation.
>
> However, we emphasize that **this case is an exception rather than the norm**. Across the majority of benchmarks, CircuitEvo consistently surpasses all baselines, owing to its ability to (1) generate high-quality initial candidates through evolutionary LLM prompting and (2) refine them via our structure-aware symbolic function optimizer. These components enable CircuitEvo to achieve robust performance gains on most circuits, even when the search space is highly complex.
>
> ### Question 4
> > **I look forward to the clarifications that address the above weaknesses.**
>
> We sincerely thank the reviewer for the constructive feedback. We greatly appreciate the time and effort spent in identifying these weaknesses. We will carefully address all the points raised, clarify the corresponding issues, and incorporate the necessary revisions to strengthen the manuscript. **Your insightful comments are invaluable in helping us improve the quality and clarity of our work.**

---

> > ### Comment · Reviewer_LswG · 2025-11-22
> >
> > Thanks for your response! I will maintain my score.

---

> ### Author Response · Authors · 2025-11-22
> **Further response to Reviewer LswG**
>
> Dear Reviewer LswG,
>
> We deeply appreciate your thoughtful evaluation of our work! Thank you for the time, care, and expertise you invested in reviewing our paper!

---

### Official Review · Reviewer_XHWa · 2025-11-03

**Soundness:** 2
**Presentation:** 3
**Contribution:** 3
**Rating:** 4
**Confidence:** 3

**Summary:**

CircuitEvo proposes an LLM-based framework for logic synthesis that iteratively evolves circuit programs toward improved compactness while preserving functional accuracy. The key innovation is a graph-structured circuit program representation that enables LLMs to understand and generate circuits through evolutionary prompting strategies. A structure-aware function optimizer ensures correctness by appending substructures based on circuit functionality completion theory. Experiments show 6.74% improvement in circuit size over state-of-the-art methods while achieving 100% accuracy across benchmarks with up to 16 inputs and 69 outputs.

**Strengths:**

1. The graph-structured program formulation (Figure 1) is genuinely clever - it encodes topological structure in a hierarchical, textual format that preserves connectivity information while being LLM-compatible. This bridges the gap between graph-based circuit representations and language-based reasoning.
2. Comprehensive evaluation across 16 circuits, 7 baselines, 3 different LLM backbones (GPT-3.5, Deepseek-V3, Qwen2.5-7B), and multiple optimization operators demonstrates robustness. The ablation study (Table 4) clearly shows each component contributes.

**Weaknesses:**

1. The "structure-aware function optimizer" uses existing LS tool (ABC) to generate subprograms Pa and Pb when the LLM-generated circuit is incorrect. This raises fundamental questions:

If ABC can generate correct subprograms, why not use ABC for the entire circuit?
How much of the final circuit comes from LLM vs. ABC?
Baselines don't receive this "rescue" mechanism - is the comparison fair?
Table 4 shows w/o Completion drops accuracy to 67-86%, suggesting LLMs contribute incomplete circuits that ABC must fix

2. Severe scalability limitations with no path forward:

Limited to 16 inputs, 69 outputs.
Shannon decomposition (Equation 1) has exponential complexity O(2^(n-1)).
Authors acknowledge "computational constraints limit... to 12-15 components" but provide no solution.
No analysis of asymptotic complexity or scaling behavior.

**Questions:**

See Weakness

---

> ### Author Response · Authors · 2025-11-22
> **Response to Reviewer XHWa---Part 1/4**
>
> Dear Reviewer XHWa,
>
> We greatly appreciate your careful reading and constructive feedback! We have provided further responses as follows. We sincerely hope that our responses could properly address all your concerns. If so, we would deeply appreciate it if you could raise your score. If not, please let us know your further concerns, and we will continue actively responding to your comments and improving our submission.
>
> ### Weakness 1.1
> > **If ABC can generate correct subprograms, why not use ABC for the entire circuit?**
>
> We sincerely thank the reviewer for the insightful question. In response, we compare our method---which uses ABC for subprogram generation---with directly applying ABC to the entire circuit. As shown in Table a, **our approach consistently produces more compact circuits than ABC alone**. This improvement is primarily enabled by two factors: (1) our **function optimization module**, which **provides a more expressive and general mapping than ABC’s built-in synthesis operators** such as SOP and BDD, and (2) the **evolutionary LLM-based generator**, which **progressively discovers higher-quality initial programs for optimization**.
>
> **The function optimization module:** the ABC’s synthesis process can be regarded as a mapping $f_{\text{ABC}} : P_{\emptyset} \rightarrow P_{\text{accurate}}$, where ABC maps only from an empty initial circuit to a functionally correct circuit. In contrast, our function-completion module implements a more general mapping  $f_{\text{HIS}} : P \rightarrow P_{\text{accurate}}$, which corrects any initial program—not just an empty circuit—into a functionally correct one. The advantages of our mapping functions lie in two aspects:
> - **(a) Generality:** ABC can only start from a fixed initialization, whereas our module can refine *arbitrary* initial circuits.
> - **(b) Mapping quality:** As shown in Table b, when both start from an empty initial circuit, the circuits produced by our module are **strictly more compact** than those synthesized by ABC.
>
> **The Evolutionary LLM-Based Generation:** Building on the function optimization module, we observe (see Figure 7 in the submission) that the compactness of the final synthesized circuit is strongly influenced by the quality of the initial circuit. This motivates the use of a domain-specific LLM generator, combined with local search, to iteratively explore the program space and produce increasingly promising initial candidates. These candidates are subsequently refined by the function optimization module, enabling continuous improvements in circuit compactness while preserving functional correctness. The evolutionary optimization results on Random4 and LogicNets2 (see Table c) further verify that **our method progressively enhances AIG compactness across iterations**, ultimately surpassing what ABC alone can achieve.

---

> ### Author Response · Authors · 2025-11-22
> **Response to Reviewer XHWa---Part 2/4**
>
> **(Continued from "Response to Reviewer XHWa---Part 1")**
>
> **Table a**: We compare our method---which leverages ABC for subprogram generation---with the baseline that directly applies ABC heuristics (SOP and BDD) to the entire circuit.
> |             | SOP       | BDD       | Best baseline | CircuitEvo (GPT-3.5-turbo) |
> | ----------- | --------- | --------- | ------------- | -------------------------- |
> | Circuit     | Init Node | Init Node | Init Node     | Init Node                  |
> | Arithmetic1 | 77        | 77        | 77            | **74**                     |
> | Arithmetic2 | 105       | 112       | 105           | **102**                    |
> | Arithmetic3 | 1424      | 1490      | 1424          | **1206**                   |
> | Espresso1   | 85        | 85        | 85            | **71**                     |
> | Espresso2   | 175       | 173       | 173           | **128**                    |
> | Espresso3   | 3302      | 3371      | 3302          | **3184**                      |
> | Espresso4   | 920       | 944       | 920           | **848**                    |
> | LogicNets1  | 148       | 135       | **135**       | 139                        |
> | LogicNets2  | 102       | 115       | 102           | **74**                     |
> | LogicNets3  | 214       | 242       | 214           | **211**                    |
> | Random1     | 42        | 42        | 42            | **41**                     |
> | Random2     | 39        | 37        | 37            | **36**                     |
> | Random3     | 124       | 110       | 110           | **60**                     |
> | Random4     | 154       | 156       | 154           | **138**                    |
> | Random5     | 190       | 130       | 130           | **105**                    |
> | Random6     | 1529      | 1514      | 1514          | **1110**                   |
> | Average     | 539.38    | 545.81    | 532.75        | **470.44**                 |
>
> **Table b:** Comparison between our function optimization module and the default ABC mapping module. All circuits are mapped starting from an empty initial circuit. The results demonstrate that our module produces noticeably more compact circuits.
> |             | ABC function mapping | Our function mapping |
> | ----------- | ------------- | ---------------------------- |
> | Circuit        | Init Node     | Init Node                    |
> | Arithmetic1 | **77**        | 79                           |
> | Arithmetic2 | **105**       | 129                          |
> | Arithmetic3 | 1424          | **1235**                     |
> | Espresso1   | 85            | **83**                       |
> | Espresso2   | 173           | **133**                      |
> | Espresso3   | 3302          | **3183**                     |
> | Espresso4   | **920**       | 1043                         |
> | LogicNets1  | **135**       | 151                          |
> | LogicNets2  | 102           | **79**                       |
> | LogicNets3  | 214           | **208**                      |
> | Random1     | **42**        | 43                           |
> | Random2     | 37            | **35**                       |
> | Random3     | 110           | **62**                           |
> | Random4     | 154           | **142**                      |
> | Random5     | 130           | **114**                      |
> | Random6     | 1514          | **1465**                     |
> | Average     | **532.75**    | **511.50**                   |
>
> **Table c:** The best-performing circuit size improves as the number of iterations increases.
> | Iterations | Random4 | LogicNets2 |
> |------------|------------|---------|
> | 1          |     151    | 77      |
> | 2          | 151        | 76      |
> | 3          | 146        | 76      |
> | 4          | 146        | 75      |
> | 5          | 146        | 75      |
> | 6          | 138        | 74      |
> | 7          | 138        | 74      |
> | 8          | 138        | 74      |
> | 9          | 138        | 74      |
> | 10         | 138        | 74      |

---

> ### Author Response · Authors · 2025-11-22
> **Response to Reviewer XHWa---Part 3/4**
>
> ### Weakness 1.2
> > **How much of the final circuit comes from LLM vs. ABC?**
>
> We sincerely thank the reviewer for the insightful question. As shown in Table d, **the average functional accuracy of the test circuits before functional optimization is 88.82%**, and **the remaining 11.18% functional gap is corrected during the functional-optimization stage**, which **introduces an average node increase of 41.44%**. Overall, the generated circuits can be categorized into two representative groups:
>
> **1. Circuits directly generated with functional correctness and compact structure:** As shown in Table d, **half of the test circuits are generated as functionally correct** and compact designs without requiring substantial intervention from the functional optimizer. This benefit stems from the strong generative capability of the LLM and our evolution-based, domain-specific prompting strategy, which effectively guides the model toward high-quality designs.
>
> **2. Circuits optimized from high-quality initial solutions:** As shown in Table d, **the best results of other half of the circuits  emerge from the high-quality initial candidates discovered by our method**. By extensively exploring the initial-solution space, our approach identifies strong starting points that substantially reduce the size increase incurred during functional optimization, ultimately leading to more compact final circuits.
>
> **Table d:** We provide the detailed generation circuits accuracy and size before function optimization.
> | Circuit     | size before optimization | acc before optimization(%) | size after optimization | acc after optimization(%) | ABC generation  rate(%) |
> | ----------- | ------------------------ | -------------------------- | ------------------------ | ------------------------- | -------------------- |
> | Arithmetic1 | 74                       | 100.00                     | 74                       | 100.00                    | 0.00                 |
> | Espresso1   | 71                       | 100.00                     | 71                       | 100.00                    | 0.00                 |
> | Espresso2   | 128                      | 100.00                     | 128                      | 100.00                    | 0.00                 |
> | LogicNets3  | 211                      | 100.00                     | 211                      | 100.00                    | 0.00                 |
> | Random1     | 41                       | 100.00                     | 41                       | 100.00                    | 0.00                 |
> | Random2     | 36                       | 100.00                     | 36                       | 100.00                    | 0.00                 |
> | Random3     | 60                       | 100.00                     | 60                       | 100.00                    | 0.00                 |
> | Random6     | 1110                     | 100.00                     | 1110                     | 100.00                    | 0.00                 |
> | Arithmetic2 | 62                       | 85.77                      | 102                      | 100.00                    | 39.22                |
> | Random5     | 49                       | 83.30                      | 105                      | 100.00                    | 53.33                |
> | Random4     | 20                       | 84.51                      | 138                      | 100.00                    | 85.51                |
> | Espresso3   | 200                      | 82.60                      | 3184                     | 100.00                    | 93.72                |
> | LogicNets2  | 3                        | 60.63                      | 74                       | 100.00                    | 95.95                |
> | Arithmetic3 | 27                       | 67.87                      | 1206                     | 100.00                    | 97.76                |
> | LogicNets1  | 2                        | 70.12                      | 139                      | 100.00                    | 98.56                |
> | Espresso4   | 9                        | 86.40                      | 848                      | 100.00                    | 98.94                |
> | **Average** | \                        | **88.82**                  | \                        | **100.00**                | **41.44**            |
>
> ### Weakness 1.3
> > **Baselines don't receive this "rescue" mechanism - is the comparison fair?**
>
> We sincerely thank the reviewer for the insightful question. We would like to clarify that **we have applied our function optimization mechanism to all inaccurate circuits generated by the baseline methods**, ensuring that every baseline output was corrected to a functionally valid circuit before comparison. Therefore, **the size comparison reported in Table 2 of the manuscript constitutes a fair evaluation across all methods**.

---

> ### Author Response · Authors · 2025-11-22
> **Response to Reviewer XHWa---Part 4/4**
>
> ### Weakness 2
> > **Severe scalability limitations with no path forward:
> Limited to 16 inputs, 69 outputs. Shannon decomposition (Equation 1) has exponential complexity O(2^(n-1)). No analysis of asymptotic complexity or scaling behavior.**
>
> We sincerely thank the reviewer for the insightful comments. Following your suggestions, we conducted a thorough experiments to analyze the scaling behavior of our methods.
>
> Specifically, we randomly generate lots of circuits by varying the number of inputs from 16 to 20 and the number of outputs from 1 to 100. The results in Table e demonstrate that **our method with GPT-3.5-turbo achieves a substantial scalability advantage over the state-of-the-art machine learning approach DNAS**. Specifically, our method can **successfully generalize to 18 inputs and 100 outputs with about 10000 nodes**. Motivated by the scalability results, we observe that **the primary limitation of our method arises from the increasing circuit size**, which leads to longer and more complex generated programs. To address this problem, we propose two potential solutions.
>
> Firstly, **leveraging large language models with substantially longer context windows** can significantly enhance the scalability of our approach. In fact, the state-of-the-art models support context lengths that are roughly 250 times larger than that of GPT-3.5-turbo. Once paired with these LLMs, our method has the potential to directly generate more complex circuits.
>
> Secondly, for very large-scale circuits---where the corresponding programs are significantly longer than the context window length---**circuit decomposition** techniques can be employed to break down the circuit into smaller, more manageable sub-circuits that can be learned or optimized individually. For instance, [1] adopts **Shannon decomposition**, which partitions the original truth table into several sub-truth tables based on input variable conditioning. Each sub-truth table is independently synthesized and then combined to reconstruct the full circuit. **Although this method has a complexity of $O(2^{\,n-1})$, the number of decompositions can be controlled to balance synthesis difficulty and computational cost.** In a different approach, [2] employs a Monte Carlo Tree Search (MCTS)-based sampling strategy, which **performs multiple rounds of input-output sampling** to construct sub-truth tables. These sub-components are subsequently synthesized and merged, enabling scalable synthesis for large and complex circuits.
>
>
> **Table e:** Comparative evaluation of our method and the state-of-the-art ML method DNAS in terms of scalability performance. Here, *NA* indicates that the method failed to generate a valid circuit within the rebuttal time limit.
> | PO=1       | PI=16     | PI=17     | PI=18     | PI=19     | PI=20     |
> | ---------- | --------- | --------- | --------- | --------- | --------- |
> | Method     | Init Nd   | Init Nd   | Init Nd   | Init Nd   | Init Nd   |
> | CircuitEvo | 436       | 626       | 901       | 1287      | 1850      |
> | DNAS       | 558       | NA        | NA        | NA        | NA        |
> | **PO=5**   | **PI=16** | **PI=17** | **PI=18** | **PI=19** | **PI=20** |
> | Method     | Init Nd   | Init Nd   | Init Nd   | Init Nd   | Init Nd   |
> | CircuitEvo | 1028      | 1444      | 2113      | 2944      | 4212      |
> | DNAS       | 1256      | NA        | NA        | NA        | NA        |
> | **PO=10**  | **PI=16** | **PI=17** | **PI=18** | **PI=19** | **PI=20** |
> | Method     | Init Nd   | Init Nd   | Init Nd   | Init Nd   | Init Nd   |
> | CircuitEvo | 1457      | 2055      | 2971      | 4188      | 5953      |
> | DNAS       | 1641      | NA        | NA        | NA        | NA        |
> | **PO=20**  | **PI=16** | **PI=17** | **PI=18** | **PI=19** | **PI=20** |
> | Method     | Init Nd   | Init Nd   | Init Nd   | Init Nd   | Init Nd   |
> | CircuitEvo | 2079      | 2939      | 4181      | 5955      | 8410      |
> | DNAS       | 2222      | NA        | NA        | NA        | NA        |
> | **PO=50**  | **PI=16** | **PI=17** | **PI=18** | **PI=19** | **PI=20** |
> | Method     | Init Nd   | Init Nd   | Init Nd   | Init Nd   | Init Nd   |
> | CircuitEvo | 3209      | 4629      | 6529      | 9292      | NA        |
> | DNAS       | 3689        | NA        | NA        | NA        | NA        |
> | **PO=100** | **PI=16** | **PI=17** | **PI=18** | **PI=19** | **PI=20** |
> | Method     | Init Nd   | Init Nd   | Init Nd   | Init Nd   | Init Nd   |
> | CircuitEvo | 4520      | 6393      | 9236      | NA        | NA        |
> | DNAS       | NA        | NA        | NA        | NA        | NA        |
>
> [1]. Wang, Zhihai, et al. Towards next-generation logic synthesis: A scalable neural circuit generation framework. NeurIPS 2024.
>
> [2]. Cheng, Shuyao, et al. "Automated CPU design by learning from input-output examples." IJCAI 2024.

---

### Meta-Review · Area_Chair_8JH5 · 2026-01-04

**Summary:**

The paper presents CircuitEvo, a framework integrating LLMs with evolutionary algorithms for logic synthesis. The key innovation is a novel graph-structured representation enabling LLMs to optimize circuit topology, paired with a structure-aware optimizer to ensure functional correctness. The authors provided a comprehensive rebuttal that resolved initial mixed reviews (8, 6, 4, 4). Key strengths confirmed during the discussion include:

1. Strong Empirical Performance: The method achieves a 6.74% average reduction in circuit size compared to state-of-the-art methods while maintaining 100% functional accuracy.

2. Rigorous Baselines: The authors successfully addressed concerns by adding comparisons to Genetic Programming (LGP) and Neural Architecture Search (T-Net/DNAS), outperforming both.

3. Scalability & Utility: New data demonstrated scalability up to 18 inputs/100 outputs, surpassing comparable ML baselines. Ablation studies further justified the hybrid approach, proving that LLMs provide high-quality initial topologies that classical solvers (like ABC) fail to discover in isolation.

**Reviewer Concerns:**

1. Fairness of Comparisons (Reviewer XHWa): The reviewer questioned if the comparison was fair since CircuitEvo uses a "structure-aware optimizer" (ABC-based) to fix errors, while baselines might not. The authors clarified that they applied the same function optimization mechanism to all inaccurate circuits generated by the baselines before measuring size. This ensures the reported improvements are due to the LLM's superior topology generation, not just the post-processing.

2. Necessity of Evolutionary Components (Reviewer 8gZq, m5UB): Reviewers questioned the contribution of the evolutionary strategy (E1/M1) versus just the LLM or the optimizer. The authors provided new ablation studies showing that removing specific evolutionary operators (like E1 or M1) degrades performance, confirming that the evolutionary cycle adds value beyond the raw LLM generation.

3. Missing Baselines (Reviewer LswG, m5UB): Reviewers requested comparisons against Genetic Programming (LGP), DARTS, and T-Net. The authors implemented a Linear Genetic Programming (LGP) baseline (since no direct competitor existed) and compared against DARTS and T-Net (clarifying T-Net is the same as their DNAS baseline). CircuitEvo outperformed all of them in accuracy and compactness.

4. Statistical Significance (Reviewer LswG): The reviewer questioned that the results were based on single runs. The authors conducted three independent runs with different seeds, reporting a very low standard deviation (0.29 nodes), demonstrating stability.

5. Dependence on ABC/Classical Tools (Reviewer XHWa): They questioned if ABC (the optimizer) is used to fix the circuit, is the LLM actually doing anything useful. The authors showed (Table d) that ~50% of circuits are generated correctly by the LLM before optimization. For the others, the optimizer fixes them. Crucially, they showed that starting with the LLM-generated topology yields significantly smaller circuits than running ABC from scratch (empty initialization), proving the LLM provides a "high-quality starting point" that classical solvers cannot find on their own.

6. LLM Scale Impact (Reviewer 8gZq): Lack of analysis on 7B vs. 14B models. Now new experiments with Qwen2.5-7B vs. 14B showed consistent performance gains with the larger model.

**Reviewer Scores:**

I think Reviewer 8gZq and XHWa would like to change their scores from 4 to 6, since most of their concerns are addressed. The other reviews might remain their positive scores.

---

### Decision · Program_Chairs · 2026-01-26

Accept (Poster)